# Pathobiology and clinical significance of malignant pleural effusions

Allison T Woods[1,2,3,8], Abner A Murray[2,3,4,8], Benjamin G Vincent [1,2,3,5✉], Jason Akulian [2,3,6✉] & Chad V Pecot [1,2,3,4,7✉]

## Abstract

**Metastatic malignant pleural effusion (MPE) represents advanced-stage cancer and is defined by the establishment of metastatic tumor foci within the pleural space. It is most commonly associated with high degrees of morbidity and mortality. Annually, over 150,000 cancer patients in the United States develop MPE, which is associated with a dismal median survival of 3–12 months. As such, efforts must be made to understand the complex biological factors driving MPE pathophysiology. In this review, we discuss what is currently known and identify knowledge gaps regarding the intrinsic MPE biology of cancer cells and the heterotypic interactions between tumor cells and the immunologic pleural ecosystem. Furthermore, we discuss the clinical opportunities of studying MPE and identify promising directions for MPE research that may lead to a deeper understanding of the disease, ultimately aiming to enhance clinical outcomes for patients with advanced cancer.**

**Keywords** Malignant Pleural Effusion; Non-small Cell Lung Cancer; Pleural Tumor Microenvironment; Tumor–Immune Interactions; Cancer Cell-intrinsic Drivers
**Subject Categories** Cancer; Respiratory System

## Introduction

The pleura is a fine, multilayered membrane that consists of the visceral pleura, which directly lines the lung parenchyma. At the hilum, this membrane reflects back onto and covers the chest wall, diaphragm, and mediastinum, becoming the parietal pleura. The pleural space is located between the visceral and parietal pleural membranes and contains small volumes (~5–15 mL) of serous fluid (D'Agostino and Edens, 2025). The pleural cavity has multiple physiological roles, acting as a lubricant and cushion for the lungs, providing augmentation of lung elastic recoil, and maintaining normal intrapleural pressure. Normally, pleural fluid represents a homeostatic, sterile microenvironment; however, exogenous factors, including infection, trauma, stress, or disease (i.e., respiratory illness, cardiovascular or pulmonary events, or cancer), can alter its composition as well as that of the pleural space/lining. The accumulation of an abnormal collection of fluid within the pleural space is called a pleural effusion. Pleural effusions are a result of disruptions within the homeostatic balance between production and/or drainage of pleural fluid. The degree of disruption required to develop a pleural effusion should not be understated, as the pleural space can drain nearly thirty times more fluid than what is typically produced (Antunes et al, 2003).

While pleural effusion can be a complication of non-cancerous events occurring in the thoracic cavity, this review focuses on metastatic malignant pleural effusions (MPE), a hallmark of advanced-stage cancer and a clinically significant endpoint in multiple malignancies. MPE are defined by recurrent accumulation of large intrapleural fluid collections that, despite systemic treatment and drainage, are ultimately associated with a high degree of morbidity and mortality (Shaikh et al, 2015). In addition, MPE differs from the normal pleural space by the presence of tumor cells, inflammation (as indicated by elevated LDH and inflammatory mediators), elevated protein levels, and typically a lymphocyte-predominant cell count (MPE may also be neutrophil-predominant as well) (Jacobs et al, 2022). MPE can occur in both the setting of primary pleural carcinoma (mesothelioma) and more commonly as metastasis from extra-pleural cancers, the most common being metastatic non-small cell lung cancer (NSCLC) and breast cancer (Fig. 1) (Dorry et al, 2021).

MPE is diagnosed by the identification of malignant cells in pleural fluid cytology (sensitivity ~60–70%) or pleural tissue biopsy, with imaging (CT and ultrasound) aiding in assessment and procedural planning. It occurs in approximately 15% of lung cancer patients at diagnosis and in up to 50% with advanced-stage disease (Kassirian et al, 2023). Other frequent causes include breast cancer, lymphoma, gastrointestinal, genitourinary, and gynecologic malignancies (Gonnelli et al, 2024). Patients commonly present with dyspnea, cough, and chest pain, though some effusions are asymptomatic and detected incidentally on imaging.

[1]Department of Cell Biology & Physiology, University of North Carolina School of Medicine, Chapel Hill, NC, USA. [2]Lineberger Comprehensive Cancer Center, University of North Carolina School of Medicine, Chapel Hill, NC, USA. [3]Carolina Center for Pleural Disease, University of North Carolina School of Medicine, Chapel Hill, NC, USA. [4]Division of Oncology, Department of Medicine, University of North Carolina School of Medicine, Chapel Hill, NC, USA. [5]Department of Microbiology and Immunology, Curriculum in Bioinformatics and Computational Biology, Computational Medicine Program, Division of Hematology, University of North Carolina School of Medicine, Chapel Hill, NC, USA. [6]Division of Pulmonary and Critical Care, Section of Interventional Pulmonology and Pulmonary Oncology, University of North Carolina School of Medicine, Chapel Hill, NC, USA. [7]RNA Discovery Center, University of North Carolina School of Medicine, Chapel Hill, NC, USA. [8]These authors contributed equally: Allison T Woods, Abner A Murray. ✉E-mail: benjamin_vincent@med.unc.edu; akulian@email.unc.edu; pecot@email.unc.edu

**Glossary**

| | | | |
|---|---|---|---|
| **17-DMAG** | HSP90 inhibitor that also suppresses IKKα/IKKβ activity in NF-κB signaling. | **CCR2/CCL2 axis** | Chemokine–receptor pathway recruiting monocytes and polarizing macrophages in malignant effusions. |
| **ANGPTL4** | VEGF-regulated glycoprotein promoting vascular permeability and pleural dissemination. | **CD163** | Scavenger receptor for hemoglobin–haptoglobin complexes; canonical M2-like macrophage marker. |
| **AXL/MERTK axis** | Pair of TAM-family receptor tyrosine kinases regulating macrophage efferocytosis, immune suppression, and tumor progression. | **CD206 (MRC1)** | Mannose receptor; endocytic receptor enriched on M2-like macrophages. |
| **BAFF-R / TACI / BCMA signaling** | TNFR superfamily pathways controlling B-cell survival and differentiation; BAFF-R (TNFRSF13C) binds BAFF, TACI (TNFRSF13B) and BCMA (TNFRSF17) bind APRIL/BAFF. | **CD40/CD40L pathway** | Co-stimulatory axis between antigen-presenting cells and T cells essential for adaptive activation. |
| | | **CD47** | "Don't-eat-me" signal; ligand for SIRPα that inhibits phagocytosis. |
| **Bevacizumab** | Recombinant humanized monoclonal antibody targeting VEGF; anti-angiogenic therapy. | **CD93** | Cell-surface receptor involved in adhesion and angiogenesis; expressed on endothelium and myeloid cells. |
| **BLZ945** | Small-molecule CSF1R inhibitor used to target CSF1R⁺ macrophages. | **CD95 (Fas)** | Death receptor; engagement by FasL triggers apoptotic signaling. |
| **Bregs (Regulatory B cells)** | Immunosuppressive B-cell subset (e.g., IL-10, TGF-β production) modulating T-cell and myeloid activity. | **Chemokines (CCL, CXCL families)** | Small secreted proteins that direct immune-cell trafficking and tumor–immune crosstalk. |
| **CAFs (Cancer-associated fibroblasts)** | Activated stromal fibroblasts that remodel ECM and shape immune infiltration and tumor growth. | **CSF1/CSF1R axis** | Growth-factor pathway regulating macrophage differentiation, proliferation, and survival. |
| **CCR2** | Chemokine receptor (cognate for CCL2/MCP-1) mediating monocyte recruitment; therapeutic target for axis blockade. | **Damage-associated molecular patterns (DAMPs)** | Endogenous danger signals (e.g., HMGB1, HSPs, ATP) released/exposed by stressed or dying cells that activate innate immunity. |

MPE fluid can be conceptualized as two main components: cellular and acellular. In healthy individuals, acellular pleural fluid is compositionally similar to that of serum. However, within pleural disease, the extent to which the presence of the acellular MPE fluid itself drives pathology versus merely being an indicator of underlying pathology is unknown. This distinction remains an open question, with mounting evidence suggesting pleural fluid can modulate immune activity and tumor behavior (Ge et al, 2024). Ultimately, the formation of effusions in malignant pleural disease is associated with worsened clinical outcomes; median survival after MPE diagnosis ranges from 4 to 7 months, varying by tumor type (Gonnelli et al, 2024; Bashour et al, 2022; Feller-Kopman and Light, 2018; Feller-Kopman et al, 2018). Patients with lung or gastrointestinal cancers have the shortest median survival (2–3 months), while those with mesothelioma or hematologic malignancies may survive up to a year (9–12 months) (Santhi et al, 2025; Bielsa et al, 2008b; Cimen et al, 2022). In NSCLC, MPE is an independent adverse prognostic factor; patients with MPE have a median overall survival of 3 months compared to 5 months in those without, and a significantly lower 1-year survival rate (12.6% vs. 24.8%) (Wang et al, 2025b; Kumar et al, 2024; Epaillard et al, 2021; Morgensztern et al, 2012). Although clinical management primarily focuses on symptomatic relief, i.e., thoracentesis, indwelling pleural catheters, or pleurodesis, the underlying cancer guides systemic therapy. The high morbidity and mortality associated with MPE underscore the need for a deeper understanding of its biology. Although the primary causes of malignant and benign pleural effusions are well characterized, ongoing research focuses on elucidating the distinct molecular and cellular mechanisms, including the contributions of acellular fluid components, that underlie the formation of MPE and their prognostic impact across various tumor types.

The pathogenesis of MPE is primarily mediated by tumor-induced vascular hyperpermeability, angiogenesis, and impaired lymphatic drainage, with tumor- and host-derived factors such as vascular endothelial growth factor (VEGF), inflammatory cytokines, and immune cell interactions playing central roles. Although specific oncogenic mutations may influence tumor behavior, current evidence suggests that specific gene mutations are not the primary drivers of MPE formation, and their direct role remains incompletely understood (Agalioti et al, 2015). Instead, MPE formation is best understood as the product of a complex and dynamic interplay between tumor cells, stromal elements, immune mediators, and the specialized physiology of the pleural cavity. Understanding how malignant cells reshape the pleural microenvironment, both at the cellular and molecular levels, will be crucial for developing novel therapeutics to enhance overall survival (OS) in patients with MPE. Beyond its clinical significance, the MPE compartment provides a unique opportunity to study tumor–immune interactions in an anatomically constrained and fluid-accessible microenvironment. Immune and stromal populations within this space are directly exposed to tumor-derived signals and may reflect or influence broader systemic immune states.

To this end, our review will highlight advancements made toward understanding MPE biology. For an overview of current treatment methods, we refer the reader to other reviews referenced (Yang and Wang, 2023; He et al, 2021; Penz et al, 2017; Donnenberg et al, 2019a; West and Lee, 2006; Işık et al, 2013;

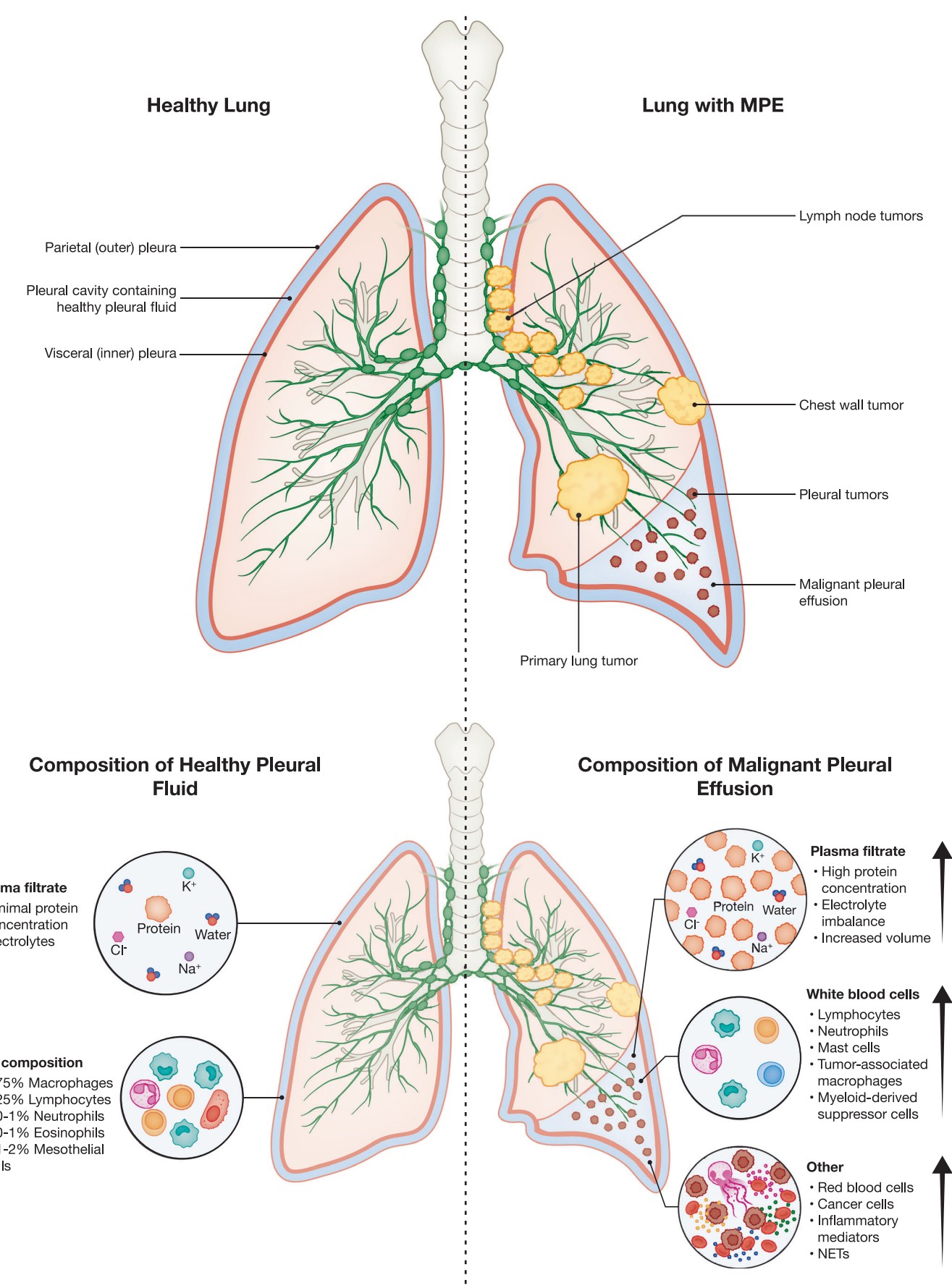

**Figure 1. Composition of a healthy pleural cavity compared to a pleural cavity with malignant pleural effusion.**

Pleural fluid in healthy individuals is a serous fluid whose cellular component is composed of mesothelial cells and monocytic leukocyte populations. This serous fluid acts as both a lubricant between the visceral and parietal pleural membranes and as a radial traction force, allowing the lungs to inflate and deflate while avoiding friction and adhesions to the chest wall. In patients with cancer, malignant cells can infiltrate the pleural cavity during metastasis. A common complication of pleural metastases is malignant pleural effusion (MPE), which arises from inflammation and lymphatic obstruction, resulting in the accumulation of protein-rich fluid within the pleural space and creating an inflamed, immunosuppressive microenvironment. The cellular component of MPE comprises a variety of cell types, including cancer cells, immune cells, and mesothelial cells, which interact with one another. The acellular fluid component contains a large number of molecular mediators, such as aberrant proteins, growth factors, and inflammatory cytokines. In healthy individuals, pleural fluid is a homeostatic, serous fluid with a cellular component composed mainly of white blood cells (~75% macrophages and ~25% lymphocytes with minimal presence (~0–2%) of neutrophils, eosinophils, and mesothelial cells) (Noppen et al, 2000; Noppen 2004). Upon transformation of healthy pleural fluid into MPE, this cellular balance shifts to contain tumor cells and becomes highly T cell dominant (lymphocytic MPE) or neutrophil dominant (neutrophilic MPE), which, as we will discuss later in this review, can affect clinical outcomes. [Note: The image illustrates differences between the healthy and malignant pleural space. These processes can occur in either hemithorax as well as bilaterally.].

Gonnelli et al, 2024; Yang et al, 2017). Here, we discuss cellular, non-cellular, and molecular drivers of MPE formation, including the roles of vascular permeability, angiogenesis, and lymphatic drainage. We highlight the methods employed by tumor cells to hijack non-malignant host cells and drive MPE and further discuss how cancer cell intrinsic drivers mediate highly pro-tumorigenic signaling cascades that initiate MPE. Additionally, we investigate how innate and adaptive immune cells contribute to the development and progression of MPE.

# Cancer cell-intrinsic drivers of MPE

Animal models of MPE have been previously described, and we have summarized prior and newer models concerning several features (Psallidas et al, 2016), including genetic context, advantages, and disadvantages (Table 1). Mechanistic studies using engineered models have linked oncogenic alterations to MPE pathogenesis, especially through the modulation of inflammatory and angiogenic signaling pathways.

## Oncogenic drivers and chemokine signaling

The most common oncogenic drivers of lung adenocarcinoma (LUAD) have also been implicated in MPE development, including *KRAS* (Agalioti et al, 2017; Marazioti et al, 2018), *EGFR* (Chiang et al, 2022; Wu et al, 2008; Tsai et al, 2015; Carter et al, 2017), *ALK* (Schwalk et al, 2021), *BRAF* (Ho et al, 2017), *MET* (Roscilli et al, 2016), *TP53* (Frille et al, 2024; Swanton and Govindan, 2016), and *PI3K* (Yin et al, 2016), among others. Histologic subtype also appears to influence the risk of MPE. Micropapillary-predominant LUAD, strongly associated with EGFR mutation, has been independently linked to pleural invasion and positive pleural cytology (Qu et al, 2025; Mikubo et al, 2018; Matsumura et al, 2016; Cao et al, 2015). Acinar-predominant tumors, while also enriched for EGFR mutations, show a more modest association with MPE formation (Dong et al, 2016). In contrast, LUAD harboring KRAS mutations has been associated with extrathoracic rather than pleural space metastasis (Dormieux et al, 2020; Lohinai et al, 2017; Renaud et al, 2016; Morales-Oyarvide and Mino-Kenudson, 2014). However, rather than merely cataloging driver mutations, we now review emerging data comparing their prevalence in MPE-positive versus MPE-negative tumors, including human genomic and epidemiologic studies. In addition to canonical oncogenic drivers, recent studies underscore the importance of pleural-specific

transcriptional reprogramming, defined as convergent adaptations in gene expression driven by the pleural microenvironment rather than the primary tumor genotype in promoting effusion formation and immune escape. These signatures reflect pleural-specific pressures, suggesting that the MPE phenotype may arise from a convergence of tumor-intrinsic drivers and niche-specific adaptations.

## KRAS-mutant tumors drive pleural immune remodeling

KRAS, one of the most frequently mutated genes in human cancers, is often associated with worsened patient outcomes and is the causative oncogene in several of the most deadly cancers, including NSCLCs, pancreatic cancers, and colorectal cancers (Huang et al, 2021a). In mouse models, KRAS-mutant cancer cells (including lung, colon, and malignant pleural mesothelioma (MPM) cancers with G12C, G12S, G12V, G13R, Q61H, and Q61R mutations) secrete high levels of CCL2 into the effusion fluid, resulting in the recruitment of peripheral blood inflammatory $CD11b^+GR1^+CCR2^+$ myeloid cells through the CCL2–CCR2 (the cognate receptor of CCL2) signaling axis. Furthermore, CCR2 null mice were protected from MPE development after direct pleural administration of 3 different $KRAS^{MUT}$ MPE-competent cancer cell lines. Furthermore, mice treated either daily with a standard drug injected intraperitoneally or only once with an encapsulated intrapleural formulation of the KRAS inhibitor deltarasin were reported to have a significant decrease in both the volume and incidence of MPE development, as well as a decrease in the number of inflammatory CCR2+ cells in the effusion fluid. In line with this, the administration of anti-CCL2 is effective in treating and controlling MPE development in mice (Agalioti et al, 2017; Marazioti et al, 2013). Studies of KRAS-driven tumors have also highlighted the upregulation of neutrophil-attracting chemokines, such as CXCL1, CXCL2, and IL-8, forming a chemokine axis that promotes vascular leakage, neutrophil extracellular traps (NETs) formation (NETosis), and the recruitment of immunosuppressive myeloid cell populations (Teijeira et al, 2020a, 2020b; Shang et al, 2020; Alfaro et al, 2016; O'Hayer et al, 2009). This inflammatory circuit likely synergizes with CCL2–CCR2 signaling and is further reinforced by tumor-derived extracellular vesicles that reprogram mesothelial cells toward a pro-angiogenic phenotype (Das et al, 2025; Kadomoto et al, 2021; Arkhypov et al, 2020).

KRAS mutations also drive the release of tumor-derived extracellular vesicles (TEVs) (Fong and Lee, 2022), which have been shown to reprogram mesothelial cells to adopt a pro-inflammatory and pro-angiogenic phenotype (Sriwastva et al,

**Table 1. Animal models for studying MPE biology.**

| Cancer of origin (cell line or genetic model) | Delivery route | Species | Immune competence | Transformation context | Advantages | Disadvantages |
|---|---|---|---|---|---|---|
| Lung Adenocarcinoma (PC14, aka PC9; PC14PE) (Yano et al, 2000b; Yeh et al, 2006) | Intravenous | Athymic Mice | Immunodeficient | EGFR$^{del19}$, p53$^{R248Q}$ | Relevant Human Genetics | No primary tumor, No T-Cells |
| Lung Adenocarcinoma (PC14, aka PC9) (Ohta et al, 2001) | Intrapleural | Rats | Immunodeficient | EGFR$^{del19}$, p53$^{R248Q}$ | Relevant Human Genetics, Larger Animal Enables Easier Catheter Placement | No primary tumor, No T-Cells |
| Lung Adenocarcinoma (Lewis Lung Carcinoma) (Merrick et al, 2019; Agalioti et al, 2017) | Intrapleural | C57/B6 Mice | Immunocompetent | KRAS$^{G12C}$, NRAS$^{Q61H}$ | T-Cell/Cancer Cell Interface; Indwelling Pleural Catheter (IPC) enables serial fluid sampling | Murine Cell Line, IPC is technically challenging |
| Lung Squamous Carcinoma (KNS-62) (Boehle et al, 2000) | Intrapleural | SCID Mice | Immunodeficient | HRAS$^{Q61L}$, p53$^{R249S}$ | Relevant Human Genetics | No primary tumor, No T- or B-Cells |
| Lung Squamous Carcinoma (SKMES-T1) (Harrison et al, 2020) | Orthotopic Lung | Athymic Mice | Immunodeficient | Carcinogen-Induced (Smoking) | Orthotopic Tumor, Selected from In Vivo Selection | No T-Cells |
| Lung Squamous Carcinoma (LN2-2 and LN4K1) (Porrello et al, 2018) | Orthotopic Lung | DBA2 Mice | Immunocompetent | Carcinogen-Induced by 3-Methylcholanthrene | Orthotopic Tumor, T-Cell/Cancer Cell Interface | Murine Cell Lines |
| Colon Adenocarcinoma (MC38) (Giannou et al, 2015) | Intrapleural | C57/B6 Mice | Immunocompetent | KRAS$^{WT}$, Mismatch Repair (MMR) Deficient | T-Cell/Cancer Cell Interface; Responsive to immunotherapies | Murine Cell Line |
| Mesothelioma (EHMES-10) (Edakuni et al, 2006) | Intrapleural | SCID Mice | Immunodeficient | RET-NCOA4 translocation | Orthotopic Tumors, Relevant Human Genetics | No T- or B-Cells |
| Fibrosarcoma (MethA) (Kimura et al, 2000) | Intrapleural | Balb/C Mice | Immunocompetent | Several p53 mutations | T-Cell/Cancer Cell Interface | Murine Cell Line |
| Mesothelioma (Conditional Genetic Model) (Jongsma et al, 2008) | Intrapleural (Adeno-Cre) | FVB/n Mice | Immunocompetent | Nf2;Ink4a or Nf2;p53 knockout | Spontaneous MPE development | Longer latency for MPE formation |

2023; Ludwig et al, 2020; Kuriyama et al, 2020). In preclinical lung tumor models, TEVs enriched in miR-5110 activated pleural mesothelial cells via the surface receptor CD93, promoting chemokine production, immune cell infiltration, and vascular leak (Zhang et al, 2024b, 2024a). These findings suggest that KRAS-driven TEV signaling represents a novel tumor-intrinsic mechanism supporting MPE pathogenesis and identify CD93 and miR-5110 as potential therapeutic targets (Zhang et al, 2024a). Zhang et al performed integrated single-cell RNA sequencing and spatial transcriptomic analyses of MPE-associated LUAD, identifying distinct epithelial cell subclusters enriched in KRAS mutations that co-expressed high levels of IL-6 (Zhang et al, 2024d). These cytokines are localized to tumor regions near exhausted CD8$^+$ T cells that co-express PD-1, LAG3, and TIM3, implicating KRAS-driven cytokine programs in the spatial orchestration of immune dysfunction within the pleural niche (Donnenberg et al, 2023). Notably, epithelial sub-clustering exhibiting high antigen presentation and TGF-β pathway activity is associated with enhanced interactions between immunosuppressive macrophages and exhausted T cells, suggesting multiple axes of tumor-intrinsic immune remodeling in MPE (Huang et al, 2021b).

KRAS-mutant and other aggressive LUADs have also been shown to exhibit tumor-intrinsic upregulation of anti-apoptotic and immune evasion mechanisms. MPE-associated tumor cells express high levels of PD-L1 and FasL, while downregulating CD95 (Fas receptor), thereby avoiding T cell- and NK cell-mediated cytotoxicity and resisting extrinsic apoptosis (Tumino et al, 2019; Sikora et al, 2004). This altered expression profile is intrinsic to the tumor cells and may contribute directly to immune evasion within the pleural cavity. Furthermore, elevated expression of glutathione peroxidase 4 (GPX4) and its stress-response regulator NUPR1 suggests a multifaceted resistance to ferroptosis as a tumor-intrinsic survival mechanism within the effusion microenvironment (Wu et al, 2023b).

Subsequent studies have reported the role of pro-inflammatory cytokine interleukin-1β (IL-1β) derived from host CCR2+ myeloid cells as a necessary culprit in driving MPE-associated inflammation, ultimately caused by plasma extravasation into the pleural space (Marazioti et al, 2018). Transcriptional profiling of $KRAS^{MUT}$ cancer cells (LUAD, colon adenocarcinoma, and MPM) revealed an increased level of *CXCL1* and *PPBP* transcripts, both of which code for chemokine ligands that bind to CXCR1+ and CXCR2+ myeloid cells, suggesting CXCL1 and/or PPBP involvement in recruitment of myeloid cell populations to the pleural space as a direct cause of $KRAS^{MUT}$ cancer cell signaling. This study demonstrated that MPE formation relies on NFκB signaling, initially triggered by myeloid-derived IL-1β. They demonstrated activation of non-canonical IκK kinase alpha (IKKα) driven NFκB signaling in $KRAS^{MUT}$ cancer cells. This was evidenced by sustained MPE formation and IKKα activity, even in the absence of the canonical NF-κB-activating kinase, IKKβ. Also reported were significant decreases in both overall MPE occurrence and volume when treating MPE-competent mouse models with combination therapy targeting both mutant KRAS (using deltarasin) and IKKα (using 17-DMAG, an inhibitor of IKKα, IKKβ, and HSP90), as compared to control groups. Interestingly, IKKα alone was insufficient in conferring MPE, as observed when treating MPE-incompetent cancer mouse models expressing wild-type *KRAS* ($KRAS^{WT}$) with exogenous IKKα; however, IKKα was necessary for

MPE competence in $KRAS^{MUT}$ MPE models, as shown by the decreased ability to confer MPE when IKKα was inhibited. Together, these data provide rationale for further study of IL-1β, KRAS, and IKKα, a previously underappreciated driver of oncogenic NF-κB signaling, as potential therapeutic targets of MPE (Marazioti et al, 2018; Stathopoulos et al, 2005). This inflammatory cascade may converge with KRAS-mediated neutrophilic chemokine signaling, further amplifying immune exclusion and vascular leak.

### EGFR-mutant tumors

*EGFR* mutations (which most frequently occur in LUAD) have also been implicated in the development of MPE. High rates of *EGFR* driver mutations have been found in LUAD-MPE, with a detection rate of approximately 70% in specific East Asian populations (Yang et al, 2017; Wu et al, 2008). One of the most common activating *EGFR* mutations ($EGFR^{L858R}$) has been shown to drive LUAD-MPE formation via the CXCL12-CXCR4 chemokine signaling axis in an ERK/MAPK-dependent manner. Increased CXCL12-CXCR4 signaling has also been shown to promote neutrophil maturation and extravasation from the bone marrow via increased tumor cell expression of CXCR4 (Tsai et al, 2015). This relationship suggests that an area of potential scientific inquiry into the role of CXCR4 in the metastatic capability associated with MPE, through chemoattraction-related migration and invasion of tumor cells, should be explored.

EGFR-mutant tumors have been reported to upregulate VEGF and IL-6, resulting in the promotion of pleural angiogenesis and the polarization of pleural macrophages toward a tumor-supportive M2-like phenotype. Single-cell and bulk transcriptomic profiling of pleural tumor cells from NSCLC and breast cancer patients with confirmed MPE have highlighted ANGPTL4 as a key effector of pleural vascular permeability and tumor transcoelomic spread. ANGPTL4, a VEGF-regulated glycoprotein, has been shown to be highly expressed in EGFR-mutant LUAD MPE samples, correlating with increased levels of pleural effusion volume, MMP9 secretion, and endothelial barrier disruption (Younis et al, 2025). These findings position ANGPTL4 as a tumor-intrinsic driver of pleural invasion, potentially synergizing with EGFR-driven VEGF and IL-6 expression to promote remodeling of pleural mesothelial cells, the stroma, and the tumor immune microenvironment (TIME). EGFR activation has also been linked to the suppression of CXCL10, reducing CD8$^+$ T cell infiltration and the impairment of immune-mediated clearance of tumors within the pleural space (Sumimoto et al, 2023; Kalinowski et al, 2014; Mascia et al, 2003). Complementary data show that EGFR-mutant MPE exhibit upregulation of the immune evasion molecule CD47, a macrophage checkpoint that interacts with SIRPα to inhibit phagocytosis. Importantly, this upregulation of CD47 appears to be tumor-intrinsic and independent of pleural immune cell composition, as confirmed by RNA-seq analysis of sorted MPE tumor cells (Hu et al, 2024). Elevated CD47 and PD-L1 expression on MPE tumor cells has been associated with reduced cytolytic activity in CD8$^+$ T cells, suggesting that EGFR-mutant tumors may exploit redundant inhibitory pathways to prevent pleural immune clearance of tumors (Zhang et al, 2025c; Yang et al, 2021). This concept is supported by recent clinical findings that cytologically confirmed MPE are associated with significantly lower PD-1 expression on CD4$^+$ T cells, and reduced co-expression of PD-1

and IL-10R on CD8$^+$ T cells, relative to benign effusions, suggesting MPE tumor cells may actively suppress checkpoint receptor induction in the effusion microenvironment (Mosleh et al, 2024). While the surface phenotype was measured on immune cells, the altered checkpoint landscape appears to be tumor-driven and distinct from T cell exhaustion, reinforcing a model in which tumor-intrinsic factors modulate the functional differentiation of effector lymphocytes within the pleural space. Transcriptomic profiling of matched primary and MPE-derived tumor cells in NSCLC revealed pleural-specific upregulation of endoplasmic reticulum (ER) stress and protein translation pathways, accompanied by a decrease in the expression of NF-κB and oxidative stress-associated gene signatures. These differences suggest that tumor cells adapt to the pleural niche by altering antigen processing, inflammatory signaling, and protein handling, possibly to evade immune detection or mitigate pleural-specific stressors (Mahmood et al, 2024). Notably, lactate accumulation, observed in both EGFR- and KRAS-driven MPE, appears to contribute not only to acidification of the pleural environment but also to metabolic crosstalk with pleural immune cells. These interactions favor norepinephrine metabolism and further macrophage suppression, potentially linking oncogenic signaling to adrenergic stress responses in the pleural niche. MPE tumor cells may actively impair T cell exhaustion by reducing antigenicity or altering immunogenic signaling. When compared to primary tumors, MPE were associated with decreased expression of exhaustion-associated markers on T cells, including PD-1, TIGIT, and CTLA-4, despite maintaining a presence of T cells. These findings suggest a tumor-intrinsic failure to engage or sustain T cell effector function, potentially due to low MHC expression or immune-evasive signaling (Mahmood et al, 2024). These findings underscore the multifaceted role of EGFR in orchestrating both tumor-intrinsic signaling and immune evasion in MPE biology. Interestingly, IL-6 and IL-8 were among the most abundant tumor-secreted factors detected across diverse cancer types in the pleural space, regardless of primary site, further implicating these cytokines as universal effectors of pleural metastasis and effusion (Donnenberg et al, 2024). Mahmood et al further demonstrated that MPE positive for malignant cells exhibited elevated expression of immunomodulatory ligands, including PD-L1, CD40, and FasL, compared to cytology-negative MPE. This supports the hypothesis that tumor-intrinsic output is a major driver of local immune dysregulation in MPE (Mahmood et al, 2024).

Consistent with these data, pleural tumor cells exhibit a conserved transcriptional program characterized by high IL6 and CXCL8 (IL-8) expression, preserved across NSCLC and breast cancer MPE (Donnenberg et al, 2024, 2023; Alexandrakis et al, 2000; Donnenberg et al, 2019b). These cytokines are tightly correlated with increased secretion of MMP2 and MMP9, promoting extracellular matrix degradation and facilitating pleural invasion (Chang et al, 2024; Fousek et al, 2021; Teixeira et al, 2016). Importantly, these findings support a model in which primary site-specific factors do not solely drive pleural metastasis but instead reflect convergent tumor-intrinsic adaptations to the pleural niche. While these mechanistic studies offer valuable insight into the effector programs downstream of KRAS and EGFR mutations, their clinical relevance must be interpreted in the context of real-world mutation prevalence and pleural tropism observed in human tumors. Despite these mechanistic findings, multiple clinical

datasets suggest that KRAS mutations are less frequent in human MPE than in MPE-negative NSCLCs (Lohinai et al, 2017). In a comparative genomic analysis of LUADs with and without cytology-proven MPE, a lower frequency of KRAS than EGFR mutations was found in the MPE-positive group, with an odds ratio significantly below 1 (0.35 [0.14–0.86]) for KRAS mutation prevalence in MPE (Smits et al, 2012).

Cell-intrinsic mechanisms ranging from the expression of immunosuppressive ligands to metabolic, epigenetic, and clonal adaptations represent promising targets for therapeutic intervention. By disrupting these programs, future strategies may reduce pleural tumor dissemination, reprogram the MPE microenvironment, and enhance responsiveness to systemic and intrapleural therapies. This discrepancy likely reflects species differences, the artificial nature of engineered mouse models, and the complex interplay of additional genetic, epigenetic, and host immune factors in human disease. While KRAS-driven mechanisms are well-established in experimental systems and provide valuable mechanistic insight into CCL2-dependent MPE formation, current human clinical data do not support a predominant role for KRAS mutations in the pathogenesis of MPE. Instead, MPE formation in humans appears to be associated with a broader range of molecular alterations, including a higher prevalence of EGFR mutations and a lower prevalence of STK11 mutations, as well as tumor-host interactions and microenvironmental drivers (Ruan et al, 2020). Emerging comparative analyses further support the reduced prevalence of STK11 mutations in MPE-positive LUAD, particularly among East Asian cohorts (Pineda et al, 2024). However, unlike STK11, alterations in KEAP1 and SMARCA4 do not appear to be less frequent in MPE-positive cases. Indeed, some studies report an enrichment of KEAP1 and SMARCA4 mutations in M1-stage tumors overall, although these were not stratified by MPE status (Gandhi et al, 2025). As such, while STK11 appears to be selectively depleted in MPE(+) LUAD, additional multivariate analyses are needed to clarify the roles of KEAP1 and SMARCA4 in pleural dissemination (Schoenfeld et al, 2020). Notably, both STK11 and KEAP1 mutations are now associated with the exclusion of CD8$^+$ T cells from the tumor microenvironment, defining subtypes of LUAD with marked immune resistance and leading to their investigation as major contributors to poor response to immunotherapy in MPE-associated tumors.

## Metabolic and epigenetic reprogramming

Recent mechanistic work also implicates tumor-derived lactate, produced through enhanced glycolysis in KRAS-mutant and other aggressive LUADs, as a key driver of MPE by altering macrophage metabolism and sustaining immune suppression within the pleural cavity. Elevated lactate concentrations in MPE fluid are associated with tumor-associated macrophages (TAM) polarization, norepinephrine biosynthesis, and downstream ERK-mediated upregulation of immunosuppressive molecules, including PD-L1 and ARG1. In parallel, tumor cells may undergo lipid metabolic reprogramming, producing fatty acid-derived immunosuppressive mediators that further shape the pleural immune milieu by skewing macrophage function and limiting antigen presentation. In parallel, tumor cells in MPE exhibit sustained expression of hypoxia-inducible factor 1-alpha (HIF-1α) and pyruvate dehydrogenase kinase 1 (PDK1), which reinforces aerobic glycolysis and supports

metabolic resilience within the hypoxic pleural space. These transcriptional programs persist ex vivo, suggesting stable epigenetic remodeling driven by MPE-associated stressors. Such adaptations likely contribute to persistent lactate accumulation, immune suppression, and enhanced TEV signaling.

These tumor metabolic signals further reinforce a non-inflamed, M2-like microenvironment, supporting pleural fluid accumulation and immune evasion. MPE-associated tumor cells also exhibit altered proliferation dynamics, which appear to vary by cancer type. Ki-67 expression in LUAD cells was notably elevated in MPE relative to matched primary tumors, suggesting that pleural dissemination selects proliferative clones in some histologic contexts (Laberiano-Fernandez et al, 2024). These findings suggest that the pleural space may promote or maintain a hyperproliferative, metabolically active subpopulation of tumor cells, particularly in lung cancer, whereas other tumor types (e.g., breast carcinoma) exhibit divergent adaptations. In addition to glycolytic reprogramming, tumor cells may also engage in altered lipid metabolism, producing immunomodulatory lipid mediators and reshaping the polarization of pleural macrophages through paracrine signaling. IL-6/sIL-6Rα signaling not only reinforces angiogenic and epithelial–mesenchymal transition (EMT)-promoting pathways within the pleura, but also contributes to CD8[+] T cell suppression, macrophage polarization, and neutrophil recruitment, highlighting its dual role as both a tumor-intrinsic driver and immunologic amplifier (Donnenberg et al, 2024). Emerging data also suggest that tumor epigenetic reprogramming, including EZH2-mediated H3K27me3 enrichment and promoter hypermethylation, may enhance immune escape and pleural tropism in select subsets of NSCLC and MPM. These epigenetic programs may converge with soluble factors and lineage programs to define the initiation of MPE. Interestingly, while PD-L1 expressions were undetectable on tumor cells in MPE, pleural macrophages exhibited increased PD-L1 positivity (Laberiano-Fernandez et al, 2024). This inverse pattern suggests a tumor-intrinsic rewiring of immune suppression away from direct engagement and toward outsourcing immunoregulation to bystander immune cells. Such a strategy could permit immune escape while minimizing tumor cell visibility to cytotoxic T lymphocytes or therapeutic checkpoint blockade. This capacity may be reinforced by tumor transcriptional plasticity, allowing dynamic shifts between epithelial and mesenchymal states. Such flexibility enhances the ability of tumor cells to invade the pleura, evade immune detection, and thrive in fluid environments. In support of these adaptations, recent primary human studies have demonstrated that LUAD tumor cells within MPE exhibit robust expression of the pro-inflammatory chemokine IL-8 and its receptor CXCR1, with co-positivity detected in 65% of cytology cell blocks from MPE patients (Chang et al, 2024). This autocrine and paracrine signaling loop enhances CXCR1 expression, sustaining tumor cell survival, migration, and EMT-like phenotypes, which promotes anchorage-independent spheroid formation in pleural fluid. Notably, IL-8 stimulation was sufficient to induce epithelial–mesenchymal transition, upregulate cancer stem cell markers, and promote invasion in both pleural-derived and canonical LUAD cell lines, underscoring its tumor-intrinsic role in pleural adaptation and anoikis resistance. These findings support IL-8/CXCR1 as a cancer cell-encoded program for pleural fluid colonization, survival, and progression. Recent single-cell analyses further support the metabolic reprogramming of pleural tumor

cells in LUAD-associated MPE. Luo et al, (2024) identified a distinct epithelial subcluster enriched for glycolytic enzymes and pro-proliferative genes, including MKI67, LDHA, and ENO1 (Luo et al, 2024). These cells also expressed high levels of the immunosuppressive molecule PD-L1 and engaged in enriched ligand–receptor crosstalk with M2-like macrophages and exhausted CD8[+] T cells. These findings reinforce the concept that tumor-intrinsic metabolic states are tightly linked to immune suppression within the pleural niche, and that pleural dissemination may select for metabolically active, immune evasive clones. Recent single-cell transcriptomic analyses have further highlighted how tumor-imposed metabolic pressures in the pleural cavity reprogram not only tumor cells but also the surrounding immune compartment. In a matched single-cell RNA-sequencing study of blood and MPE samples from NSCLC patients, Yang et al identified transcriptional upregulation of glycolysis, cysteine metabolism, methionine cycling, and alpha-linolenic acid pathways across multiple immune cell types within the MPE, including CD8[+] and CD4[+] T cells, B cells, and NK cells. These immune populations expressed elevated levels of metabolic effectors such as HK2, PKM, and PFK, which are classically associated with hypoxia adaptation, effector dysfunction, and cellular exhaustion. This metabolic rewiring was distinct from that observed in circulating immune cells from the same patients, reinforcing the idea that cancer cell-driven nutrient depletion and metabolite accumulation within the pleural fluid actively constrain immune cell function and promote local immunosuppression. Such findings underscore the breadth of metabolic remodeling induced by tumor cell colonization of the pleural space, with consequences not only for tumor cell survival but for the functional silencing of anti-tumor immunity (Huang et al, 2021b).

## Tumor-derived cytokines and immune-modulatory factors

### IL-6 axis
Beyond chemokines, tumor-derived cytokines such as IL-6 and soluble IL-6 receptor alpha (sIL-6Rα) have emerged as shared features across epithelial malignancies metastatic to the pleura. In a recent proteomic analysis of 254 MPE from lung, breast, gastrointestinal, and other cancers, Donnenberg et al identified a conserved secretomic signature dominated by IL-6 axis signaling, supporting the central role of tumor-intrinsic inflammatory programs in MPE biology (Donnenberg et al, 2024). Consistent with an IL-6-driven T-cell defect, Zhang et al showed that CD8[+] T cells in NSCLC exhibit features of exhaustion, that exogenous IL-6 further increases PD-1 on patient CD8[+] cells, and that combined IL-6 and PD-1 blockade restores CD8[+] effector function (including assays using pleural-effusion-derived T cells) and improves antitumor activity in vivo (Zhang et al, 2025a). Recent work by Cheng et al further underscores the tumor-intrinsic role of IL-6 in regulating immune checkpoints within the pleural microenvironment. Using preclinical MPE models and paired patient samples, the authors demonstrated that IL-6 stimulation directly upregulates PD-L1 expression on tumor cells via STAT3 activation, thereby establishing a cytokine-mediated feedback loop that reinforces pleural immune suppression (Cheng et al, 2025). These findings suggest that the IL-6/STAT3 axis operates not only as a marker of inflammation but as a functional driver of immune evasion through checkpoint upregulation. Complementing these findings, Huang

et al (2021b) demonstrated that tumor-intrinsic metabolic and cytokine cues within MPE actively reprogram pleural immune cells at the transcriptional level. Using single-cell RNA sequencing of matched peripheral blood and MPE samples, the authors identified pleural-infiltrating T cells, B cells, and NK cells with marked upregulation of glycolysis, cysteine and methionine metabolism, and the alpha-linolenic acid pathway, a shift not observed in their circulating counterparts. These changes included increased expression of metabolic regulators such as HK2, PKM, and PFK, consistent with metabolic exhaustion and adaptation to hypoxic conditions (Huang et al, 2021b). These data suggest that cancer cells in the pleural space impose metabolic constraints that impair immune function, reinforcing the immunosuppressive niche through metabolic competition and nutrient deprivation.

### IL-8 and the CXCR1 signaling loop

Primary human studies have identified the IL-8/CXCR1 axis as a critical autocrine and paracrine signaling loop in LUAD-associated MPE. In cytology blocks from 17 patients with malignant effusions, 65% demonstrated strong co-expression of IL-8 and CXCR1 by immunohistochemistry, with significantly elevated IL-8 levels in pleural fluid compared to serum or standard culture conditions (Chang et al, 2024). Functionally, IL-8 stimulation of MPE-derived and LUAD cell lines induced epithelial–mesenchymal transition, enhanced spheroid formation, promoted the expression of cancer stem cell markers, and increased invasion and migration, suggesting a pleural-intrinsic program of tumor survival and dissemination. These findings implicate IL-8 not only as a soluble immune-modulatory cytokine but also as a tumor-intrinsic effector of plasticity, stemness, and resistance to anoikis within the fluid-filled pleural environment. Chemotherapy-stressed tumor cells can further amplify this axis by releasing tumor-derived microparticles (TMPs) enriched in IL-8, CXCL1, and CXCL2, which recruit and prolong the survival of neutrophils through CXCR1/2-dependent chemotaxis (Xu et al, 2020). In preclinical models, this TMP-driven loop enhances neutrophil-mediated MMP-9 release, NETosis, and pro-angiogenic signaling, creating a microenvironment conducive to pleural fluid accumulation and tumor progression.

### TGF-β and osteopontin

In addition to IL-8, other tumor-intrinsic cytokines such as transforming growth factor-β (TGF-β) and osteopontin (OPN) play central roles in shaping the malignant pleural effusion microenvironment. TGF-β promotes EMT, metabolic reprogramming, and resistance to apoptosis through canonical SMAD2/3 and non-canonical PI3K/AKT/mTOR signaling (Stathopoulos and Kalomenidis, 2012). These multifunctional mediators promote vascular leak, fibrosis, and pleural dissemination through both autocrine and paracrine mechanisms. Recent transcriptomic profiling of pleural tumor cells suggests that autocrine OPN and TGF-β signaling play a role in anchorage-independent survival, EMT, and resistance to fluid shear stress. OPN (also known as secreted phosphoprotein-1, SPP1) is overexpressed in MPE and provokes vascular hyperpermeability by directly disrupting endothelial junctions and inducing actin polymerization, a process partly VEGF-independent (Psallidas et al, 2013; Moschos et al, 2009). OPN also enhances angiogenesis by upregulating VEGF in pleural mesothelial and endothelial cells, while simultaneously supporting tumor cell survival and dissemination by activating NF-

κB and inhibiting apoptosis. Knockdown of tumor-derived OPN significantly reduces MPE volume without affecting primary tumor growth (Cui et al, 2009). TGF-β, in particular, drives metabolic and mesenchymal reprogramming of cancer cells, reinforcing EMT and resistance to apoptosis in the pleural environment (Miyashita et al, 2021; Hua et al, 2020; Hao et al, 2019; Bollrath and Greten, 2009). In MPM, soluble isoforms of TGF-β1, TGF-β2, and TGF-β3 are enriched in pleural fluid (Okita et al, 2024). Notably, elevated soluble TGF-β2 may help distinguish malignant from benign effusions, while increased levels of TGF-β1 and TGF-β3 correlate with inferior survival, highlighting the clinical relevance of this immunosuppressive axis. Recent single-cell transcriptomic analyses (Ran et al, 2023) identified epithelial tumor subclusters in LUAD-associated MPE with elevated expression of TGF-β pathway genes and stress-adaptation programs, supporting the tumor-intrinsic role of TGF-β signaling in pleural dissemination. Mechanistically, OPN interacts with integrins (αvβ3, αvβ5) and CD44 to promote adhesion, migration, and extracellular matrix remodeling, complementing TGF-β-driven EMT and pleural colonization (Psallidas et al, 2013)).

### TNF-α

Tumor necrosis factor-α (TNF-α) is a key pro-inflammatory and pro-angiogenic cytokine that plays a central role in the pathogenesis of MPE. Both tumor cells and host mesothelial and immune cells contribute to TNF-α levels in the pleural space, but tumor-derived TNF-α is a critical driver of effusion formation, vascular hyperpermeability, and intrapleural tumor progression. In pre-clinical LUAD models, neutralization of TNF-α markedly reduces pleural fluid accumulation and tumor dissemination, underscoring its functional importance as a tumor-intrinsic mediator of MPE (Stathopoulos et al, 2007).

Mechanistically, TNF-α acts through both autocrine and paracrine signaling loops to induce VEGF, disrupt endothelial barrier integrity, and amplify angiogenesis. Beyond its role in vascular leak, TNF-α synergizes with TGF-β to promote EMT, matrix remodeling, and pleural fibrosis, contributing to tumor invasion and effusion formation (Dash et al, 2021). TNF-α also activates NF-κB and STAT3 pathways in cancer cells, linking inflammation with tumor cell survival, proliferation, and resistance to stress within the pleural cavity (Aggarwal et al, 2009; Bollrath and Greten, 2009).

Clinical and translational data consistently demonstrate elevated TNF-α in MPE. Studies by Alexandrakis et al (2000), Odeh et al (2000), and Xirouchaki et al (2002) show that pleural fluid TNF-α concentrations are significantly higher than in serum, with pleural-to-serum ratios notably elevated in malignant versus benign effusions (Xirouchaki et al, 2002; Alexandrakis et al, 2000; Odeh et al, 2000). While local TNF-α production is not exclusive to malignancy, the magnitude of elevation and the concurrent induction of VEGF and other pro-inflammatory mediators point to a tumor-associated source. Modern proteomic studies confirmed that TNF-α is among the most abundant cytokines secreted by tumor cells in NSCLC-associated MPE (Donnenberg et al, 2019a).

Cytokine profiling of MPE has revealed that TNF-α levels often correlate with those of TGF-β and OPN, reflecting a coordinated network that amplifies inflammation, angiogenesis, and EMT. Importantly, elevated TNF-α and TGF-β are associated with pleural loculation, increased fibrosis, and impaired fibrinolysis (Chung

et al, 2005). These observations underscore the context-dependent interplay between TNF-α and other tumor-derived cytokines in shaping the pleural tumor microenvironment.

### Integrins and matrix remodeling

These pathways are in part intrinsic to tumor cell programming and may converge with canonical driver mutations such as EGFR, KRAS, and PI3K. In parallel, tumor–mesothelial interactions are often mediated by tumor-expressed integrins (e.g., ITGB1, ITGA5) and matrix metalloproteinases (MMP-2, MMP-9), which facilitate pleural seeding and mesothelial clearance, key events occurring early in MPE pathogenesis. In addition to integrin–mesothelial interactions, LUAD tumor cells in MPE induce a mesothelial-to-mesenchymal transition characterized by the upregulation of mesenchymal markers, including α-SMA, ICAM1, SERPINE1, and VEGFB, which further supports pleural invasion and fibrosis (Wu et al, 2023b). In parallel, tumor-intrinsic apoptosis has emerged as a critical upstream trigger of immune-mediated matrix remodeling. Zhao et al demonstrated that apoptotic debris from tumor cells initiates an efferocytosis cascade in pleural macrophages, leading to the secretion of IL-10. This, in turn, programs dendritic cells to secrete tissue inhibitor of metalloproteinases 1 (TIMP1), which promotes pleural vascular permeability and extracellular matrix disruption. Genetic deletion of AXL/MERTK, IL-10, or TIMP1 abrogated MPE formation in vivo, positioning tumor-driven apoptotic turnover as a key initiator of immune-stromal remodeling within the pleural niche (Zhao et al, 2021).

### CSF1–CSF1R axis and tumor–stroma crosstalk

Tumor-derived CSF1 orchestrates pleural immune suppression and vascular remodeling by activating the CSF1R axis on both TAMs and cancer-associated fibroblasts (CAFs). In a murine LUAD model of MPE, deletion of CSF1R or pharmacologic inhibition (BLZ945) reduced pleural effusion volume, macrophage recruitment, and tumor burden (Kosti et al, 2022). Importantly, tumor-derived CSF1 activated CSF1R+ CAFs, which in turn secreted CXCL2 (MIP2) to attract myeloid cells, creating a tumor–stroma–myeloid circuit that drives MPE formation. This non-redundant tumor-intrinsic cytokine pathway represents a converence point for vascular leak, mesenchymal reprogramming, and immune evasion, reinforcing the role of tumor–stromal crosstalk in MPE pathogenesis.

### Soluble immune checkpoints and decoy receptors

Emerging evidence also suggests that tumor-derived soluble CD40 may act as a decoy receptor, competitively binding CD40L and thus preventing its interaction with membrane-bound CD40 on T cells. This mechanism suppresses T cell activation and facilitates immune evasion in the pleural space, adding to the repertoire of tumor-intrinsic immunosuppressive strategies in MPE. Such decoy receptor signaling underscores how tumor-secreted proteins can disrupt critical co-stimulatory interactions and further suppress effective anti-tumor immunity in the pleural cavity. The emergence of PD-1⁻ IL-10R⁻ CD8⁺ T cell populations in MPE, as identified by Mosleh et al, (2024), further supports the idea that tumor-derived suppressive cues shape the immunophenotype of local lymphocytes, potentially reflecting a tumor-intrinsic mechanism that prevents immune recognition by turning off both checkpoint and cytokine receptor signaling axes (Mosleh et al, 2024).

Additionally, tumor cells in MPE, including NSCLC and MPM, can secrete soluble immune checkpoint molecules such as sPD-1, sPD-L1, and sCTLA-4 into the pleural fluid (Okita et al, 2024). These soluble isoforms may function as competitive inhibitors or immune decoys, blunting effector T cell activation. Intriguingly, sPD-1 levels in pleural fluid have been associated with a favorable response to anti-PD-1 immunotherapy in MPM, suggesting that some tumor-intrinsic secreted factors may paradoxically enhance therapeutic susceptibility. This paradoxical association suggests that soluble checkpoints in MPE may serve not only as biomarkers, but also as dynamic regulators of immune tone. Importantly, sPD-L1 levels may reflect tumor burden and protease activity rather than direct membrane expression, which complicates their interpretation regarding immune checkpoint inhibitor (ICI) efficacy. Targeting checkpoint shedding pathways (e.g., via ADAM10/17 inhibitors) may represent a novel adjunctive strategy to restore immune recognition. Comparative analyses of matched primary and pleural tumors also suggest subclonal selection in the pleural space, favoring anchorage-independent growth, mesenchymal transition, and low-adhesion phenotypes, features that support survival in suspension and immune evasion (Wu et al, 2023b; Seo et al, 2022; Giarnieri et al, 2013; Chen et al, 2013). Notably, malignant cells within MPE may undergo downregulation of checkpoint ligands as an adaptive immune evasion strategy. In a multiplex immunofluorescence analysis of LUAD-associated MPE and matched primary tumors, PD-L1 expression was absent in all pleural tumor cells, despite being detectable in 27% of the corresponding primaries (Laberiano-Fernandez et al, 2024). This reduction in surface checkpoint expression suggests a tumor-intrinsic reprogramming during pleural dissemination, potentially reflecting a reliance on non-redundant mechanisms of immune suppression, including the secretion of immunosuppressive factors or the manipulation of the surrounding immune microenvironment. In addition to secreting immune-modulating proteins, tumors in MPE appear capable of altering the expression of key immunologic receptors on infiltrating T cells. Reduced PD-1 and IL-10R expression on CD8⁺ T cells in MPE may reflect tumor-intrinsic efforts to render the local immune response inert by limiting receptor-ligand engagement altogether (Mosleh et al, 2024). This strategy could represent a parallel mechanism of immune evasion to those involving decoy ligands and checkpoint ligands, especially in PD-L1ˡᵒʷ tumors.

In addition to T-cell-focused checkpoints, a myeloid "don't eat me" axis operates in the pleural niche: mesothelioma cells upregulate CD47, which engages SIRPα on pleural macrophages to inhibit phagocytosis and dampen antigen presentation. This anti-phagocytic checkpoint likely contributes to the low-immunogenic phenotype of MPE and offers a complementary target to T-cell ICIs (Murthy et al, 2019). Beyond classical immune checkpoints and soluble decoys, tumor-intrinsic activation of the complement cascade has emerged as an additional axis of immune evasion in MPE. Recent transcriptional profiling of LUAD cells isolated from MPE identified significant upregulation of complement component C5 within the tumor epithelial compartment (Wu et al, 2023b). Elevated tumor-derived C5 expression was associated with diminished overall survival among patients harboring EGFR-mutant tumors, suggesting that complement signaling is a non-redundant, oncogene-linked mechanism of immune suppression. These findings position C5 as a tumor-intrinsic effector with dual

relevance, both as a prognostic biomarker and as a potential therapeutic target to disrupt pleural-specific complement–immune crosstalk.

### Exosomal immune modulators: LRG1 and beyond

In addition to soluble ligands and decoy receptors, tumor cells can deliver immunosuppressive signals via exosome-mediated packaging. A recent study in MPM identified leucine-rich α-2-glycoprotein 1 (LRG1) as a tumor-intrinsic factor secreted via exosomes that promotes macrophage polarization toward an M2-like phenotype (Wang et al, 2024a). MPM cell lines (MSTO-211H, H2452) were shown to secrete LRG1 in exosomal form, which induced increased expression of CD206, ARG1, IL-10, and TGF-β in THP-1-derived macrophages, while reducing the M1 marker CD86. These effects were abolished by knockdown of LRG1 in tumor cells, confirming the protein's tumor-intrinsic role. Complementary mechanistic data further show that LRG1's macrophage-skewing activity is time- and dose-dependent, accompanied by Smad2 phosphorylation, and attenuated by TGF-β receptor blockade (SB431542), implicating a TGF-βR/Smad2-dependent pathway in M2 programming (Wang et al, 2024a). Not all mesothelioma lines elicit equivalent macrophage skewing, highlighting tumor-secretome heterogeneity and suggesting that LRG1-driven reprogramming may vary across clones (Wang et al, 2024a). This mechanism reveals a packaging-based immune evasive strategy distinct from canonical PD-L1 or sPD-1 pathways, enabling tumors to modulate pleural macrophage phenotypes at a distance. Together, these findings position LRG1 as an upstream input that converges with the pleural TGF-β/IL-10 circuitry discussed in the TAMs section, while the exosomal route provides an additional means of long-range macrophage conditioning within the pleural space (Osorio-Antonio et al, 2025). Beyond the nanoscale exosomes typified by LRG1 packaging, chemotherapy-treated tumor cells shed larger vesicular structures, tumor-derived microparticles (TMPs), that act as mobile carriers of pro-inflammatory and immune-modulatory cargo (Xu et al, 2020). These TMPs deliver chemokines, damage-associated molecular patterns (DAMPs), and proteases that can remotely condition myeloid cells, promoting neutrophil recruitment, survival, and NETosis, and indirectly fostering a pleural microenvironment that is permissive to tumor growth and fluid accumulation. This highlights that tumor-derived vesicles across the exosome–microparticle spectrum can shape MPE biology through size- and cargo-dependent mechanisms.

## Discordant genomics and spatial heterogeneity in MPE

### Small cell, squamous, and LUAD variants in MPE formation

Beyond spatial divergence, tumor-intrinsic drivers of MPE also differ by histologic subtype. In squamous NSCLC, STK11/KEAP1 co-mutations may promote a hypoinflammatory, metabolically dysregulated pleural phenotype, while small cell MPE may exhibit neuroendocrine reprogramming marked by NEUROD1 upregulation and TP53/RB1 loss (Takumida et al, 2025; Chen et al, 2025a; Lissa et al, 2022; Swanton and Govindan, 2016). These distinctions reflect lineage-specific epigenetic and transcriptomic wiring, which may influence the composition of MPE fluid, immunogenicity, and clinical trajectory. Using single-cell mass cytometry, a feasibility study in NSCLC reported that within LUAD-associated MPE, cytokeratin-positive tumor cells span diverse EMT/MET phenotypic states rather than a single program, and that in pleural effusions the CD33+ myeloid fraction positively correlates with the CK+ epithelial fraction (Karacosta et al, 2023). This suggests that even within a single histologic class, pleural-invading clones exhibit transcriptional diversity and are tailored to shape the immune milieu.

The role of other genetic drivers, such as *BRAF, MET, TP53*, and *PI3KCA* that have been associated with MPE formation remains to be explored; preliminary data suggest that MET overexpression (Akamatsu et al, 2014), TP53 loss-of-function (Lee et al, 2004), rare BRAF mutations (e.g., V600E) (Bharti et al, 2025), and activation of the PI3K/AKT/mTOR pathway may contribute to tumor invasion, immune modulation, or epithelial–mesenchymal transition in the pleural space (Yin et al, 2016). While these alterations occur less frequently than EGFR or KRAS mutations, they represent additional contributors to MPE biology and warrant further investigation.

An intriguing clinical juxtaposition is 'wet' versus 'dry' malignant pleural disease. Pleural metastases can occur as either dry pleural dissemination (DPD) or wet pleural dissemination (WPD) (Shim et al, 2006). DPD occurs when there are pleural metastases but no presence of effusion fluid, while WPD occurs in patients who have an accumulation of pleural fluid (Johnston, 1985). When comparing patients with pathologically confirmed pleural dissemination with or without pleural effusions, those with DPD had nearly a threefold better median survival time (~38 months) compared with those with WPD (~13 months), resulting in approximately a 70% lower hazard death rate (Kim et al, 2011). This differential survival signal suggests that the intrinsic difference in tumor genetics, molecular, and immunologic correlates of WPD versus DPD warrant further study.

### Discordant genomics in matched primary vs. pleural tumors

Transcriptomic and genomic profiling of MPE has revealed that tumors within the pleural space often diverge from their matched primary tumors in terms of immune infiltration, cytokine expression, and representation of driver mutations. This includes marked upregulation of IL-8/CXCR1 expression in pleural tumor cells compared to matched primary LUADs, suggesting microenvironmental reprogramming that promotes EMT and spheroid survival (Chang et al, 2024). These spatial differences suggest that MPE sampling offers unique insights into immune escape mechanisms and treatment resistance, distinct from biopsy data obtained from other metastatic or primary sites. As such, pleural-based tumor sampling and cell-free DNA analyses may emerge as essential components of real-time, tumor-informed decision-making in patients with MPE.

Importantly, recent studies highlight intrapatient heterogeneity between primary tumors and matched pleural metastases. In NSCLC, discordance in mutation profiles between the primary lung lesion and pleural metastases has been observed in 8–16% of cases, particularly for targetable alterations like EGFR (Lee et al, 2019a; Wang and Wang, 2015; Han et al, 2011). These adaptations are likely driven by the unique selective pressures of the pleural space, which favor anchorage-independent growth. Tumor cells in MPE must survive in a fluid-based, non-adherent microenvironment,

favoring traits such as anoikis resistance, cytoskeletal remodeling, and detachment-mediated survival signaling. These discrepancies are attributed to spatial sampling bias, temporal clonal evolution, and selective immune pressures within the pleural cavity (Moghaddam et al, 2024; Swanton and Govindan, 2016), with implications for diagnostic biopsy strategies and personalized therapy selection (Lee et al, 2023; Wang et al, 2019b).

Complementing these genomic and phenotypic discrepancies, recent studies have identified elevated expression of chromatin-modifying enzymes, most notably EZH2 and KDM6B, in MPE tumor cells compared to matched primary lung tumors. These epigenetic regulators are associated with transcriptional repression of pro-inflammatory and immune-stimulatory gene networks and may drive pleura-specific immune evasion programs. Such alterations may underlie the emergence of niche-adapted tumor phenotypes distinct from those in the primary site, reinforcing the role of MPE as epigenetically distinct reservoirs of therapeutic resistance.

Recent prospective data from Tu et al reinforce the prevalence and clinical relevance of discordant genomics in MPE. In a multicenter cohort of 101 treatment-naive patients with Stage IV LUAD, comprehensive NGS of cell-free DNA from MPE supernatants revealed actionable driver mutations, including EGFR, KRAS, MET, ERBB2, and ALK, in 73.3% of cases (Tu et al, 2022). Notably, EGFR mutations were detected in 63% of MPE cell free DNA (cfDNA) samples, and MET exon 14 skipping alterations were identified in 8.9%, surpassing expected plasma detection rates. Furthermore, discordance between MPE-derived cfDNA and matched tissue was observed in 19.8% of cases, including therapeutically actionable events missed in biopsy specimens. These findings support the notion that pleural-based sampling may capture spatially enriched clones or subclonal heterogeneity overlooked by traditional tissue biopsies.

### Tumor-driven spatial patterning of T cell proximity

Emerging spatial analyses suggest that tumor-intrinsic programming may influence the localization of immune cells and immune priming before pleural dissemination. In matched primary tumors from patients who developed MPE, closer proximity between malignant cells and CD3[+], CD8[+], or PD-1[+] T cells was associated with shorter intervals to effusion formation (Laberiano-Fernandez et al, 2024). This spatial convergence suggests that tumors predisposed to pleural spread may actively structure immune architecture in ways that accelerate effusion development. Tumor-driven immune positioning could serve as a parallel or precursor mechanism to cytokine signaling, shaping the future pleural immune microenvironment through early engagement in the primary site. These observations imply that pleural dissemination is not merely stochastic but may be preconditioned by tumor-intrinsic immune choreography that structures local infiltration patterns to facilitate future immune escape.

## MPE as a diagnostic and immunophenotyping resource

Clinically, the increasing use of pleural fluid for genomic profiling offers a minimally invasive alternative to tissue biopsy (Mahmood et al, 2023). cfDNA and supernatant RNA from MPE samples demonstrate high concordance (87–93%) with tissue-based sequencing for actionable mutations, often surpassing plasma cfDNA in

sensitivity (Wang et al, 2025a; Tu et al, 2022; Son et al, 2020; Tong et al, 2019). The utility of cfDNA from MPE was further validated by Lee et al, who demonstrated that cfDNA from the acellular supernatant of MPE outperformed plasma in detecting oncogenic drivers in treatment-naive LUAD (Lee et al, 2023). In this prospective multicenter study, MPE cfDNA had a 97% success rate for NGS and enabled the detection of multiple therapeutically actionable mutations with high fidelity. Notably, the study emphasized that cfDNA from MPE may serve as a superior substrate for liquid biopsies when tumor tissue is insufficient or inaccessible and may reflect spatial heterogeneity with clinical implications for therapy selection. These data elevate MPE-derived cfDNA as a frontline diagnostic tool for molecular profiling, particularly in patients presenting with de novo malignant effusion.

Notably, supernatant RNA enables transcriptomic profiling even in acellular effusions, expanding the clinical utility of MPE beyond DNA-based mutation detection to include dynamic gene expression analysis. This positions MPE as not only a site of immune modulation but also a viable substrate for serial molecular monitoring in advanced LUAD (Wu et al, 2022). In addition to cfDNA and supernatant RNA, Mahmood et al demonstrated the utility of flow cytometry and single-cell RNA sequencing of MPE-derived tumor cells as a means to study their immunomodulatory phenotypes. As such, MPE are not only a platform for non-invasive diagnostics, but also for real-time tumor immunophenotyping in the metastatic setting (Mahmood et al, 2024). The high resolution of scRNA-seq, together with complementary multimodal assays, also facilitates immunophenotyping of MPE at a systems level. Tumor scRNA-seq was combined with patient-matched PBMC CITE-seq and TCR clonotype tracing to link tumor-intrinsic programs to immune niches (Giotti et al, 2024). This multimodal framework resolved malignant programs (e.g., proliferation, EMT, hypoxia/angiogenesis) alongside associated immune landscapes; while shown in primary pleural mesothelioma, it offers a methodological template for pleural-fluid studies. These tools offer a powerful platform for stratifying patients and tailoring pleural-targeted immunotherapies.

## Synthesis: tumor-intrinsic programs shape the MPE tumor immune microenvironment (TIME)

Together, these studies suggest that oncogenic drivers implicated in MPE, such as mutant KRAS and EGFR, do not act in isolation but modulate the immune composition of the pleural space through chemokine and cytokine signaling. KRAS-driven tumors, for instance, promote MPE through CCL2-dependent recruitment of CCR2[+] myeloid cells and IL-1β-mediated NF-κB activation, both of which enhance inflammatory infiltration and vascular leakage. Similarly, EGFR mutations upregulate CXCR4, enhancing CXCL12-mediated neutrophil trafficking and metastatic behavior. These pathways demonstrate how tumor-intrinsic alterations reshape the TIME by directing the recruitment, polarization, and function of immune cells. Moreover, metabolic reprogramming of cancer cells, especially increased glycolytic flux and lactate production, emerge as a key cell-intrinsic mechanism driving immune suppression in MPE. Lactate-induced norepinephrine biosynthesis in tumor-associated macrophages reinforces ERK signaling and PD-L1 expression, contributing to a metabolically enforced immune-suppressive state within the pleural cavity. Taken

together, these data emphasize that tumor cells are not passive residents of the pleural space. Instead, they actively shape the immune and biochemical properties of the MPE through transcriptional reprogramming, altered cytokine and ligand production, and reduced immunogenic signaling. These cancer cell-intrinsic adaptations reinforce a suppressive, non-inflammatory pleural niche, with implications for resistance to immune checkpoint blockade and localized therapeutic strategies (Mahmood et al, 2024). As will be detailed in the following sections, these intrinsic programs not only promote effusion formation but also dictate the immunologic tone of the pleural cavity, shaping T cell exclusion, neutrophil enrichment, and macrophage polarization in ways that influence both natural history and therapeutic vulnerability. The following section focuses on the cellular and molecular composition of the MPE-associated TIME and how distinct immune states influence prognosis and therapy response.

## MPE represent a highly complex TIME

During MPE establishment, cancer cells induce several microenvironmental transformations, remodeling a previously bland pleural space into one characterized by leukocyte expansion (lymphocyte and/or neutrophil predominance), immune suppression, hypoxia, and metabolic dysregulation (Aujayeb, 2022; Thomas et al, 2016). This evolving immune milieu is composed of both adaptive and innate cells, which collectively shape effusion development and tumor progression (Huang et al, 2021b; Tumino et al, 2019; Wu et al, 2018). Unlike the consistencies observed between samples of normal pleural fluid or peripheral blood, MPE do not maintain a consistent cellular makeup between patients, as some may have tumor cell-predominant MPE or immune cell-predominant MPE, which can be either lymphocyte- or monocyte-rich (Dhupar et al, 2020). Regardless, this inflammatory and immunosuppressive environment is thought to be associated with dampened anti-tumor responses, vascular hyperpermeability, and cancer cell immune evasion, all of which have been shown to correlate positively with poor disease outcomes (Desai and Lee, 2017). Understanding the cross-talk between immune cells, tumor cells, and acellular mediators within the fluid itself is crucial for comprehending the complexity of this complication (Fig. 2).

Immune profiling by Wu et al (2022) stratified LUAD-associated malignant pleural effusions (LUAD-MPE) into four immune landscape clusters (C1–C4), demonstrating that patients with "hot" adaptive immunity (C1/C2) had significantly longer overall survival compared to those with "cold" adaptive immunity (C3/C4). Hot clusters were enriched for adaptive immune cell infiltrates (primarily T cells), whereas cold clusters exhibited reduced adaptive immunity, higher expression of immunosuppressive molecules such as PD-L1 and B7-H3, and upregulation of pro-angiogenic factors, including VEGFA. Among these groups, C4 (adaptive−, innate+) was associated with the poorest survival outcomes, while C2 (adaptive+, innate−) showed the most favorable prognosis. Triggering receptor expressed on myeloid cells 2 (TREM2), a marker linked to immunosuppressive macrophage phenotypes, was the only molecule preferentially expressed in short-survival patients.

In contrast, long-survival patients exhibited preferential expression of several tumor-suppressing genes within the pattern

recognition receptor (PRR) and type 1 interferon signaling pathways (Murthy et al, 2019; Wu et al, 2022). While these findings highlight the importance of adaptive and innate immune cell interplay in shaping the immunosuppressive and tumorigenic nature of the MPE TIME, there is no evidence that "cold" tumors preferentially develop MPE or that immune landscape alone determines effusion formation. Similarly, the predictive value of hot/cold classifications for ICI responsiveness in MPE remains unproven, as survival benefits with ICIs are more likely driven by systemic tumor control than by direct modulation of the effusion.

Collectively, these data underscore that the immune architecture of MPE, while prognostic, is shaped by a dynamic interplay between adaptive and innate cells, soluble mediators, and tumor-derived factors rather than hot/cold status alone. To better understand this complexity, the following subsections examine the individual contributions of key immune cell populations beginning with innate immune players such as TAMs, mast cells, and neutrophils then exploring adaptive subsets including B cells, cytotoxic T cells, regulatory T cells, and CD4$^+$ lineages.

### Innate immune cells in MPE

#### Tumor-associated macrophages (TAMs)

MPE TAMs are present at high frequencies with ontogeny shaping their function. Monocyte-derived "small" pleural macrophages constitute the dominant M2-like TAM pool that supports tumor growth, whereas tissue-resident "large" pleural macrophages may contribute to antitumor memory (Wu et al, 2023a, 2020b). Across pleural effusions, macrophages comprise over half of the cells in the compartment, underscoring their centrality to pleural immune tone (Murthy et al, 2019). Single-cell RNA sequencing of LUAD-MPE has further resolved this ontogeny into two predominant TAM end-states: interferon-primed IFN-TAMs (IFITM3$^+$/CASP4$^+$) with relatively preserved antigen-presenting/phagocytic capacity, and lipid-associated LA-TAMs (APOE$^+$/GPNMB$^+$) enriched in MPE that display reduced APC/phagocytosis scores and a transcriptional skew toward M2-like immunosuppression (Wu et al, 2023b). Pseudo-time analyses reveal that both states originate from a monocyte-derived source, with the upregulation of ZNF331 and NUPR1 along the LA-TAM branch, indicating stress and metabolic regulatory points in pleural macrophage polarization. A smaller "proliferative macrophage" subset (TOP2A$^+$, MKI67$^+$) is also detectable, potentially representing a transitional state or self-renewing pool within the pleural niche. Functional scoring reveals a clear gradient in antigen presentation/phagocytosis: IFN-TAMs > proliferative macrophages > LA-TAMs. TAM burden may also vary by tumor histology as shown in a matched multiplex immunofluorescence (mIF) pilot wherein CD68$^+$ macrophages predominated in breast carcinoma (BC) MPE, whereas LUAD MPE were comparatively T cell-skewed (Laberiano-Fernandez et al, 2024). Despite such histology-related differences, large-scale pleural secretomics across epithelial MPE ($n > 250$) demonstrates a conserved high-abundance cytokine/chemokine milieu, supporting shared TAM recruitment and polarization cues across cancers metastatic to the pleura (Donnenberg et al, 2024). In NSCLC-MPE specifically, single-cell/spatial analyses identified recurrent-prone CLDN4$^+$ epithelial subclusters as the dominant senders of VEGFA and MIF signals to SPP1$^+$/CD74$^+$ macrophages, positioning TAMs as key recipients and amplifiers of tumor-derived permeability/

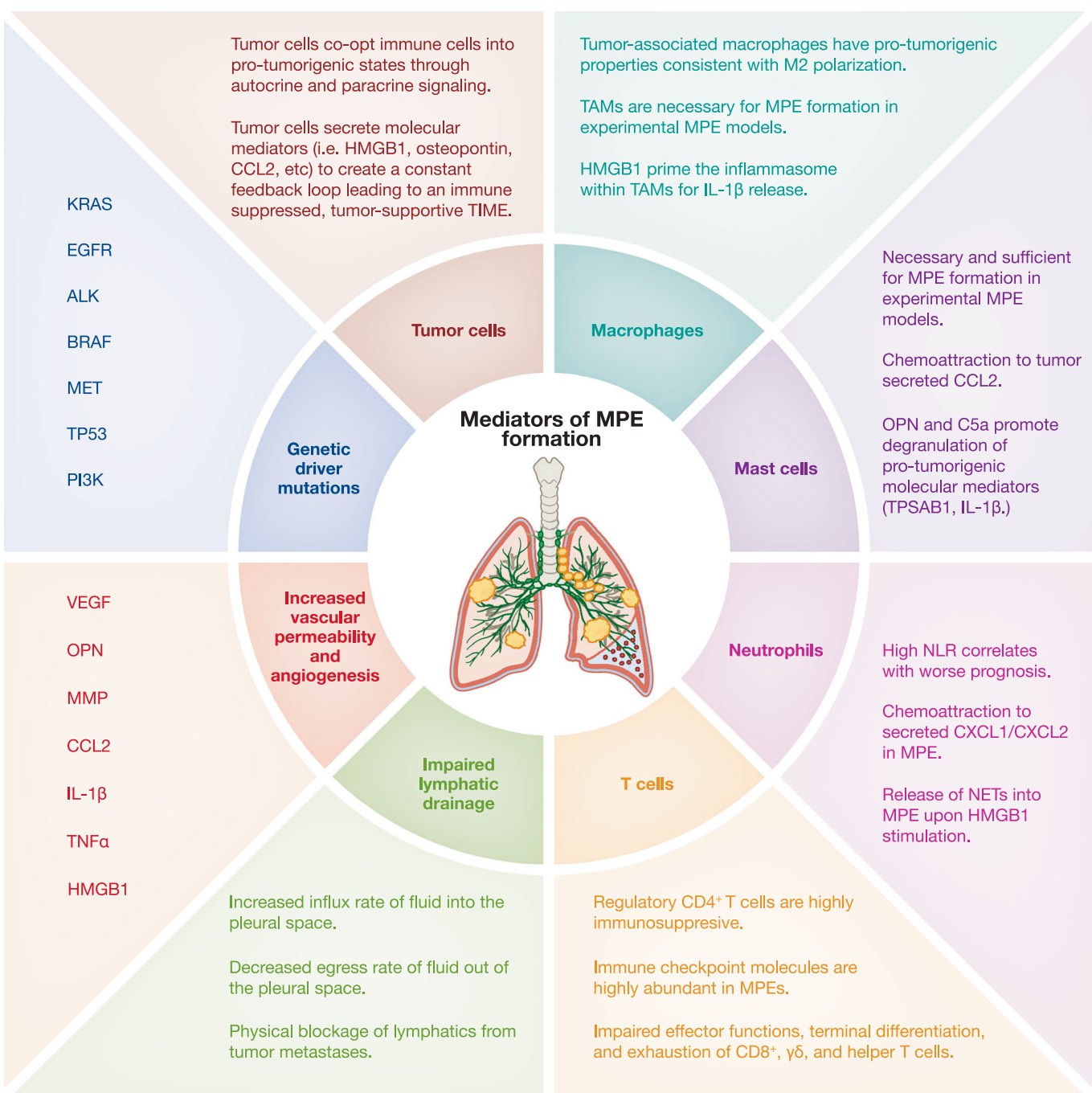

**Figure 2.  Key mediators of MPE formation.**

MPE formation reflects the convergence of tumor-intrinsic, immune, acellular, and physical drivers. Oncogenic mutations (*KRAS, EGFR, ALK, BRAF, MET, TP53, PI3K*) promote secretion of mediators including HMGB1, CCL2, IL-6, IL-8, VEGF, TNF-α, and exosomal immune modulators that remodel the pleural tumor–immune microenvironment. Tumor-associated macrophages (TAMs) polarized toward an M2 phenotype secrete IL-10, TGF-β, VEGF, HMGB1, and C5a, fueling immunosuppression, vascular leak, and inflammasome-driven IL-1β production, while mast cells, recruited via CCL2 and activated by osteopontin, release tryptase and IL-1β to further promote permeability. Neutrophils, chemoattracted by CXCL1, IL-8, and HMGB1, generate neutrophil extracellular traps (NETs), with high neutrophil-to-lymphocyte ratios (NLR) predicting poor outcomes. In contrast, T cells display exhaustion and checkpoint upregulation (PD-L1, B7-H3), regulatory CD4+ T cells are enriched and suppressive, while B cells contribute to antigen presentation and have been linked to improved prognosis. Physical and acellular drivers, including VEGF-, TNF-α-, and osteopontin-mediated angiogenesis, matrix remodeling, and impaired lymphatic drainage, further disrupt pleural homeostasis. Together, these processes sustain effusion accumulation, establish an immunosuppressed TIME, and carry distinct prognostic implications.

angiogenic and CD74–MIF immunoregulatory inputs within the pleural niche (Zhang et al, 2024d).

Within this spectrum, M2-skewed transcriptional states often enriched for CD163/CD206 are associated with the suppression of anti-tumor immunity and inferior overall survival in MPM (Laberiano-Fernandez et al, 2023; Kosti et al, 2022; Napoli et al, 2021; Yang et al, 2015b). In lung cancer-associated pleural effusions, the fraction of TNF-α-producing CD14$^+$ monocytes/macrophages was found to be markedly reduced compared to benign inflammatory effusions (e.g., tuberculosis), reflecting a shift away from anti-tumor activity and toward an immunosuppressive state (Lopez-Gonzalez et al, 2007). This myeloid-dominant, immunosuppressive signature has been confirmed by recent single-cell and deconvolution analyses in LUAD MPE (Bruschini et al, 2022; Huang et al, 2021b). Pleural DAMP–TLR cues likely contribute to this skew: HMGB1 and surfactant protein A (SP-A) are elevated in NSCLC-MPE, and SP-A positively associates with M2-polarized macrophages expressing TLR2/TLR4 (Kaczmarek et al, 2018). Notably, in the same pilot cohort, PD-L1 expression was undetectable on malignant cells in MPE cytology blocks despite being present in a subset of matched primary tumors, suggesting that pleural immunosuppression in at least some cases may rely more heavily on myeloid programs than on tumor-cell PD-L1. In matched primary tumors from the same series, CD68$^+$ macrophages were in closer proximity to malignant cells in BC than in LADC, where CD3$^+$ T cells predominated in nearest-neighbor analyses; shorter malignant cell–immune cell distances, particularly for T-cell subsets, associated with shorter MPE-free survival (Laberiano-Fernandez et al, 2024).

TAMs in MPE secrete immunosuppressive cytokines such as TGF-β and IL-10, which stimulate FOXP3$^+$ regulatory T cells and inhibit effector T cell activity (Zhao et al, 2021; Li et al, 2016). They also engage in defined feedback loops, such as CCL22-mediated recruitment of Tregs that produce IL-8, which in turn reinforces TAM TGF-β production and immunosuppression (Wang et al, 2019a). Tumor-derived LRG1, reported in soluble and exosomal forms, can further bias pleural macrophages toward M2 polarization via TGF-β receptor/Smad2 signaling (Wang et al, 2024a). Consistent with these loops, secretomic profiling of epithelial MPE quantified eleven mediators consistently ≥10 pM, CXCL10/IP-10 (~672 pM), CCL2/MCP-1 (~563 pM), sIL-6Rα (~403 pM), IL-6 (~138 pM), CXCL1/GRO (~80 pM), TGF-β1 (~77 pM), CCL22/MDC (~55 pM), IL-8 (~29 pM), CCL11/eotaxin (~13 pM), IL-10 (~11 pM), and G-CSF (~11 pM), providing quantitative support for robust monocyte recruitment (CCL2) and M2-skewing of TAMs (IL-6/sIL-6Rα, TGF-β1, IL-10) within the pleural space (Donnenberg et al, 2024). Moreover, a geometric mean sIL-6Rα:IL-6 molar ratio of ~2.7 implies pervasive IL-6 trans-signaling (IL-6 bound to soluble IL-6Rα signaling on gp130$^+$ cells) across gp130$^+$ pleural cells, including macrophages, within the pleura's contained, "bioreactor-like" environment, positioning the IL-6 axis as an upstream coordinator that cooperates with TGF-β1/IL-10 and CCL22 to sustain TAM-mediated immunosuppression (Donnenberg et al, 2024). In mesothelioma, upregulation of the anti-phagocytic checkpoint CD47 on tumor cells provides a "don't-eat-me" signal that inhibits macrophage-mediated clearance, as detailed in the Soluble Immune Checkpoints and Decoy Receptors section (Schürch et al, 2018; Tong and Wang, 2018). In contrast to infectious tuberculous pleural effusions (TPE), MPE typically show low pleural IL-32; monocytes/macrophages are the principal pleural source and IL-32 expression is IFN-γ-inducible, while IL-32γ drives TNF-α in TPE, together indicating that the pro-inflammatory IFN-γ → IL-32 → TNF-α macrophage circuit is attenuated in MPE (Wang et al, 2024c). Pleural-compartment regulation of IL-32 isoforms has also been described (↑IL-32α/β/γ and higher β:γ, α:γ ratios in pleural mononuclear cells vs blood), further underscoring distinct macrophage programming in MPE relative to TPE (Wang et al, 2024c). Complementing these contrasts, a 42-study meta-analysis found that pleural IFN-γ and TNF-α are markedly higher in TPE than in MPE (standardized mean differences ~3.30 and 2.22, respectively), while IL-4, IL-10, and IL-2 show no significant differences between etiologies (small, non-significant trends toward higher levels in MPE). Notably, pleural IL-6 was also higher in TPE than in MPE (pooled SMD 3.53), indicating that although IL-6 can be abundant in MPE, its elevation is not unique relative to infectious effusions supporting a model in which MPE TAMs experience relative Th1 cue paucity rather than uniform Th2 dominance (Zeng et al, 2022). Preclinical work in an intrapleural MPE model further implicates an IL-10 → S100A9 axis in maintaining pleural immunosuppression. In IL-10$^{-/-}$ mice, MPE volume, pleural tumor mass, vascular permeability, and total pleural macrophage burden were all reduced, accompanied by a shift from MHC-II$^-$/CD206$^+$ M2 toward MHC-II$^+$/CD206$^-$ M1 TAMs; bone marrow-derived macrophages mirrored this polarization pattern (↑iNOS, ↑IL-12β; ↓Arg1, ↓CD206). Transcriptomic profiling identified S100A9 as a key IL-10-responsive effector in MPE TAMs, and in vivo S100A9 knockdown reproduced M2 → M1 repolarization, suppressed MPE formation, and lowered pleural macrophage numbers. Functionally, reprogrammed TAMs from IL-10-deficient or S100A9-silenced conditions reduced tumor cell migration and increased tumor apoptosis in co-culture. These findings position IL-10-driven S100A9 expression as a macrophage-intrinsic mechanism promoting M2-skewed polarization and MPE pathogenesis (Pei et al, 2023). Emerging data also identify a macrophage adrenergic–metabolic axis in MPE, wherein elevated pleural lactate drives phenylalanine/tyrosine/norepinephrine (NE) metabolism in TAMs, inducing an immunosuppressive phenotype (increased PD-L1, increased ARG1) via β-adrenergic, ERK-dependent signaling. In a small observational cohort of MPE patients ($n = 6$) receiving propranolol for cardiovascular indications, CT imaging revealed a reduction in effusion volume, accompanied by a decrease in TAM frequency and PD-L1/ARG1 expression, alongside improved NK/T-cell function. In an intrapleural LLC MPE mouse model, β-blockade similarly reduced MPE volume and pleural tumor burden, repolarized TAMs toward TNF-α$^+$ states, and diminished the suppressive programs of myeloid-derived suppressor cell (MDSC). These findings identify β-adrenergic/ERK signaling as a macrophage-intrinsic node that sustains pleural immunosuppression and highlight lactate-driven catecholamine metabolism as a distinct, targetable pathway in MPE TAM biology (Zhang et al, 2025b).

In LUAD-MPE, LA-TAMs are further transcriptionally enriched for cholesterol/lipid metabolism pathways, iron-handling programs (HAMP and altered SLC40A1/ferroportin expression), and complement/coagulation cascades (including FGA, FGB, FGG), potentially linking metabolic reprogramming to immune evasion and pleural fluid accumulation. Pleural fluid profiling demonstrates broad complement activation, including C1q, C2, C4/C4b, MBL,

factor B, C3/C3a/C3b, C5/C5a, factor H, and factor I, with higher C5 levels correlating with worse overall survival, specifically in EGFR-mutant LUAD-MPE (Wu et al, 2023b). Notably, recent genetic and biochemical data indicate that pleural C1q can exert macrophage-intrinsic effects that are at least partly classical-complement-independent (with pleural C3/C4 lower than blood in murine MPE), suggesting that C1q-driven TAM programming in the pleura does not require full downstream cascade activation (Yi et al, 2024). Mechanistic interventional data further demonstrate that pleural TAMs are tractable targets in vivo. Intrapleural phosphatidylserine-coated, PEGylated liposomes carrying a STING agonist (LNP-CDN) preferentially accumulate in the pleural space and are taken up by CD11c$^+$ pleural phagocytes (monocytes/macrophages > dendritic cells), overcoming high soluble ENPP1 that degrades free CDN and enabling cytosolic STING activation. scRNA-seq and flow cytometry show TAM repolarization from M2 to M1 with gains in antigen-processing/presentation programs and iNOS↑/Arg1↓, alongside increased type I IFNs, IFN-γ, IL-2, IL-12, and IL-15/IL-15R in pleural fluid, with effects observed in both effusions and pleural tumors and reproduced ex vivo in human NSCLC MPE (Liu et al, 2022). Building on these insights into TAM plasticity, recent studies have identified a distinct subset of C1q$^+$ TAMs exhibiting fatty acid-driven metabolic reprogramming that actively suppresses T-cell function and contributes to immune checkpoint blockade resistance in MPE (Zhang et al, 2023a). Extending this, macrophage-specific or global deletion of C1qa in murine intrapleural MPE reduced pleural effusion volume and tumor burden and prolonged survival, while single-cell profiling showed contraction of M2-like macrophage states and restoration of CD8$^+$ T-cell/NK-cell effector programs (↓PD-1/LAG-3/TIM-3; ↑IFN-γ/TNF-α). Ligand–receptor analyses further identified a C1q-dependent chemokine axis whereby C1q$^+$ TAMs reinforce CCL2/CCL7/CCL8 → CCR2 signaling toward NK and T cells; macrophage C1q loss dampened this communication and relieved lymphocyte dysfunction. Functionally, targeting this network via CCR2 antagonism (RS504393) or by elevating the metabolite hippuric acid each suppressed MPE/tumor burden, with an additive benefit when combined, nominal TAM-centric strategies that warrant further study (Yi et al, 2024).

Consistent with their contribution to an immunosuppressive milieu, higher M2-polarized macrophage density and spatial proximity to tumor cells in MPM correlate with impaired anti-tumor immune responses and worse patient survival (Wang et al, 2015, 2021). Monocyte-derived macrophages are actively recruited from the circulation into the pleural cavity by tumor-derived chemokines (e.g., CCL2), where they differentiate into immunosuppressive TAMs (Chéné et al, 2016) and secrete pro-angiogenic factors (e.g., VEGF) and proteases that increase vascular permeability, contributing to fluid build-up in the pleural space. FN1 upregulation in LA-TAMs suggests potential crosstalk with reactive mesothelial cells in ECM remodeling (integrin/TNC/ICAM1 details are covered in the "Integrins and Matrix Remodeling" subsection). TAMs also release chemokines (CCL22, IL-8) along with other immunosuppressive factors that reinforce T-cell suppression and maintain a tolerogenic environment within the effusion (Wang et al, 2019a).

Given their central role in MPE pathogenesis, monocytes and macrophages in MPE are emerging as potential therapeutic targets. In preclinical models, blocking CSF-1/CSF-1R signaling, including

with the clinically relevant CSF1R inhibitor BLZ945, significantly reduced pleural fluid volume, angiogenesis, and tumor burden (Kosti et al, 2022). Similarly, targeting macrophage efferocytosis receptors (MERTK/AXL) in preclinical MPE models to interrupt the IL-10/TIMP-1 cascade that fuels MPE development; both genetic deletion and pharmacologic inhibition in these models reduced MPE progression (Zhao et al, 2021). TLR4-biased macrophage re-education has also shown proof-of-concept activity: the bacterial preparation PA-MSHA repolarized M2 → M1 macrophages in vitro, and this effect was abrogated by TLR4 blockade, supporting the feasibility of TLR4-mediated TAM reprogramming in pleural disease (Wang et al, 2015). Macrophage-derived exosomes containing miR-4443 have been shown to promote Treg differentiation in MPE, highlighting a potential translational biomarker and therapeutic target (Shao et al, 2023a). IHC validation in pleural tumors confirms the presence of CD68$^+$APOE$^+$ and CD68$^+$ZNF331$^+$ macrophages, underscoring the translational relevance of these LA-TAM markers in MPE (Wu et al, 2023b). Consistent with a macrophage-centric mechanism, intrapleural STING agonism via LNP-CDN reduced MPE volume and pleural tumor burden and, when combined with anti-PD-L1, significantly prolonged survival in wild-type but not STING$^{-/-}$ mice; detailed effects on other immune compartments and dosing/PK are discussed in their respective subsections. Preclinical safety studies reported no liver enzyme abnormalities or major organ pathology with intrapleural LNP-CDN ± anti-PD-L1 (Liu et al, 2022).

MPE TAMs may also support malignant cell proliferation by inducing pro-inflammatory signaling cascades (Kaczmarek and Sikora, 2012). Some studies of MPE formation have examined inflammasome-mediated effects on tumorigenesis and immune system surveillance. In MPM, asbestos/HMGB1–TLR4 signaling primes macrophage inflammasomes, enhancing pro-IL-1β processing to active IL-1β (Mossman et al, 2013). IL-1β mediates pro-tumorigenic effects by increasing the expression of downstream cancer stem cell markers, specifically promoting spheroid formation and CD26 (DPP4) expression in mesothelioma cells. This occurred via the interaction of TAM-secreted IL-1β with its cognate receptor IL-1R, which was previously shown to be highly expressed on MPM tumor cells compared to their non-malignant mesothelial cell counterparts and directly correlated with poorer OS (Kadariya et al, 2016). Complement C5a→IL-1β signaling remains speculative in MPE and is discussed in the inflammasome subsection; however, it is mechanistically plausible given other tumor contexts.

Together, these data position pleural TAMs as metabolically and cytokine-programmed hubs, integrating IL-6 trans-signaling, TGF-β/IL-10 and C1q axes, tumor-derived VEGFA/MIF inputs, and adrenergic cues to drive immune suppression, vascular permeability, and MPE maintenance, while offering multiple, mechanistically distinct points for therapeutic reprogramming.

### Mast cell involvement in MPE

Mast cells (c-KIT+) are key facilitators of MPE formation, with tumor-secreted CCL2 identified as a key chemoattractant for mast cells into the pleural space, and tumor-secreted osteopontin (OPN/SPP1) promoting subsequent mast cell degranulation. Quantitative pleural secretomics demonstrate a high-abundance chemokine/cytokine milieu, CCL2/MCP-1, IL-6/sIL-6Rα, IL-8, and CXCL1, consistent with a "bioreactor-like" pleural compartment that sustains mast-cell recruitment and activation (Donnenberg et al,

2024). Upon degranulation, several molecular mediators are released, including tryptase AB1 (TPSAB1) and IL-1β, which induce endothelial leak (via protease-activated receptor-2, PAR-2) and NF-κB activation in tumor cells (Fig. 3, purple). Concomitant release of histamine and VEGF further amplifies pleural vascular leak and angiogenesis (Komi and Redegeld, 2020). Complement anaphylatoxins, particularly C5a, are potent mast-cell activators and are elevated in MPE, linking the pleural complement signature to mast-cell-driven permeability and C5aR1 signaling on mast cells (Luo et al, 2022b). Enhanced mesothelial–tumor crosstalk signals in the pleural niche (e.g., tenascin-C, ICAM-1) provide plausible upstream triggers that can prime mast cells for activation, and pleural MIF elevations nominate a CD74-centered immunoregulatory pathway interfacing with mast cells (Wu et al, 2023b). In matched human samples, mast cells were significantly enriched in lung tumor biopsies compared with paired pleural effusions ($p < 0.01$), indicating that effusion-only profiling likely underestimates mast-cell burden and that mast-cell-mediated permeability/angiogenic programs localize primarily to the pleural tumor interface (Mahmood et al, 2024). Upon intrapleural coadministration of traditionally MPE-incompetent cancer cell lines with bone marrow-derived mast cells, mice developed MPE. Conversely, MPE-competent cancer cell lines were unable to form MPE in mast cell-deficient mouse models; however, MPE formation was restored in mast cell-deficient mice upon coadministration of mast cells and cancer cells, together demonstrating the necessity and sufficiency of mast cells for MPE formation. Consistent human data show increased mast-cell abundance in MPE compared with benign effusions (Giannou et al, 2015). Experiments comparing mast cell necessity within MPE formation to that of highly abundant CCL2-attracted macrophages revealed mast cells to be equally important in MPE formation, as both mast cell-deficient and macrophage-deficient mouse models resulted in the same degree of decreased effusion formation from administration of MPE-competent cells. Finally, the administration of a clinically available mast cell inhibitor, imatinib mesylate (via c-KIT inhibition), decreased effusion formation and reduced the accumulation of mast cells in murine MPE models (Giannou et al, 2015). Collectively, these data position c-KIT[+] mast cells as non-redundant drivers of pleural hyperpermeability and maintenance of MPE, suggesting testable strategies at the KIT, complement/C5a, and PAR-2 nodes.

### High neutrophil abundance in MPE is associated with worsened patient outcomes

Significant progress has been made in elucidating the role of neutrophils in cancer progression (Quail et al, 2022; Chen et al, 2022; Cui et al, 2021; Xiao et al, 2021; Teijeira et al, 2020b; Jung et al, 2019; Snoderly et al, 2019; Lee et al, 2019c; Lerman and Hammes, 2018; Nicolás-Ávila et al, 2017; Erpenbeck and Schön, 2017). In LUAD-MPE, single-cell transcriptomic profiling has identified abundant macrophage-derived CXCL1, CXCL2, and CXCL8/IL-8 programs that are canonical CXCR2-ligand drivers of neutrophil recruitment (Huang et al, 2021b). The prognostic association of this chemokine signature with poor survival in LUAD supports a model in which mononuclear cell programs help establish and sustain neutrophil-enriched pleural microenvironments. Notably, the technical exclusion of granulocytes in such atlases highlights a persistent data gap in high-resolution mapping of pleural neutrophil states, underscoring the need for granulocyte-

inclusive single-cell or CyTOF approaches. Immunohistochemical and transcriptomic profiling of MPE leukocyte populations has further shown that a higher MPO[+] neutrophil-to-CD45[+] leukocyte ratio within pleural cell blocks correlates with significantly shorter overall survival across tumor types, whereas a higher B-cell fraction is associated with improved outcomes (Wu et al, 2019). In addition to these pleural fluid-based metrics, systemic inflammatory indices such as the blood neutrophil to lymphocyte ratio (NLR) have demonstrated high predictive accuracy for survival stratification in MPE (Peng et al, 2022). In a 191-patient cohort, high-risk LENT scores (driven in part by elevated NLR) identified patients with a median survival of only ~33 days compared to ~662 days for low-risk, outperforming ECOG alone for early mortality prediction (Gayaf et al, 2021; Clive et al, 2014). Prospective data from Ozkan et al further reinforce this relationship, showing that a serum NLR ≥ 6.81, alongside elevated PLR (platelet to lymphocyte ratio) (≥ 275.08) and CRP (≥51.2 mg/L), independently stratified patients into markedly worse survival groups (AUCs ~0.69–0.71) even when infection and non-malignant effusions were excluded. These data, underscore the robustness of neutrophil-skewed systemic inflammation as an adverse prognostic signal in MPE (Ozkan et al, 2024). More recently, Suárez-Antelo et al developed and prospectively validated the GASENT score, which incorporates NLR alongside age, sex, ECOG-PS, and tumor type, and demonstrated superior survival prediction compared to LENT across 1-, 3-, and 6-month intervals (e.g., 1-month AUC 0.777 vs. 0.737). This further supports the centrality of neutrophil-driven systemic inflammation as a reproducible and clinically actionable prognostic determinant in metastatic MPE (Suárez-Antelo et al, 2025). Beyond survival, serum NLR also carries procedural relevance: in a 331-patient NSCLC-MPE cohort managed with indwelling pleural catheters, a serum NLR ≥ 2.68 independently predicted failure of spontaneous pleurodesis within 60 days, and inclusion of serum NLR in a validated nomogram improved prediction accuracy (AUC 0.745 development; 0.720 validation), linking neutrophil-driven systemic inflammation to reduced likelihood of durable effusion control (Tan et al, 2025). This reinforces the prognostic relevance of neutrophil-driven inflammation across both local and systemic compartments.

Although neutrophil predominance is commonly associated with intrapleural infection, neutrophilic MPE have been reported to make up ~20% of those sampled in a small patient cohort of MPE derived from different cancer types (Lee et al, 2019b). In a prospective analysis of malignant and benign effusions, Popowicz et al found that pleural fluid contained far fewer neutrophils than blood (median 9% vs 73% of leukocytes) and a markedly lower neutrophil–lymphocyte ratio (NLR; median 0.20 vs 4.9), with only moderate correlation between compartments. A pleural NLR > 0.745 or pleural neutrophil proportion >4.74% independently predicted significantly shorter survival, whereas pleural lymphocyte proportion was not prognostic (Popowicz et al, 2021; Akturk et al, 2016). Notably, in high-ADA pleural effusions, Shimoda et al reported that neutrophil predominance, especially when accompanied by elevated pleural WBC and CRP, was far more likely to indicate pleural infection than malignancy, whereas MPE in this setting was more often supported by pleural amylase ≥75 U/L and an ADA/total protein ratio <14 (Shimoda et al, 2022). Similarly, in a large cytologic series, histiocyte-predominant effusions (≥50% histiocytes) were most often malignant (~51%), particularly in LUAD, underscoring that MPE can present with diverse leukocyte

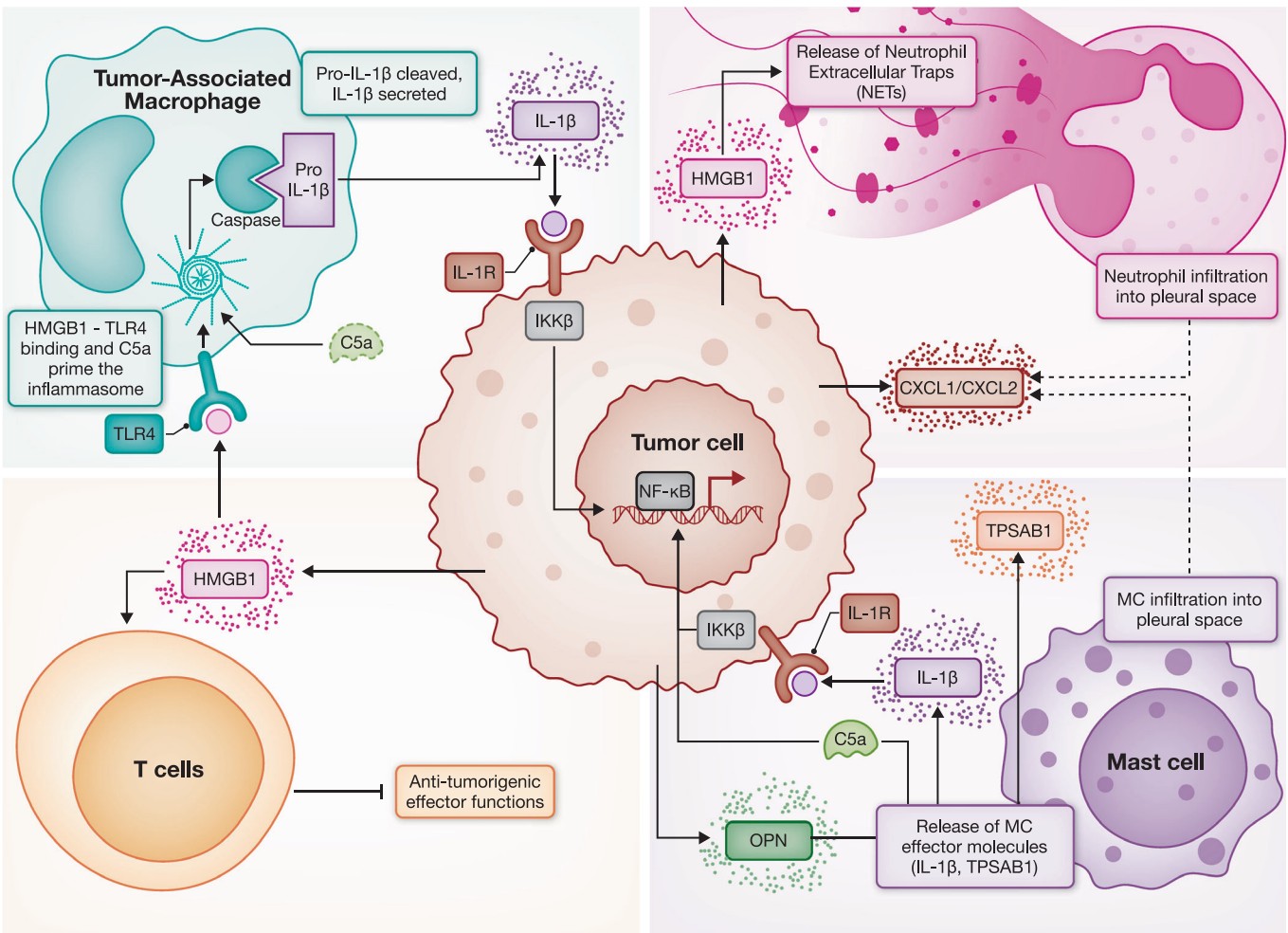

**Figure 3.  Tumor cells within the MPE polarize immune cells toward pro-tumorigenic functions.**

This schematic illustrates how tumor cells co-opt immune subsets within the pleural space to create and maintain an immunosuppressive, fluid-accumulating microenvironment. Tumor cells secrete CCL2 to recruit monocytes/macrophages and mast cells, and release HMGB1, which activates TLR4/inflammasome signaling in macrophages, driving IL-1β production. M2-polarized tumor-associated macrophages amplify vascular permeability and immune suppression through IL-10, TGF-β, and angiogenic mediators. Mast cells, recruited by CCL2 and activated by tumor-derived osteopontin, degranulate to release tryptase and IL-1β, further enhancing vascular leak. Neutrophils are recruited into the pleural space, where they generate neutrophil extracellular traps (NETs) that promote tumor progression and correlate with poor prognosis. The T cell compartment is characterized by exhausted CD8+ cells with diminished cytotoxicity and abundant regulatory T cells that enforce immune suppression. Together, these interconnected paracrine and autocrine loops reinforce a pro-tumorigenic pleural tumor–immune microenvironment, facilitating both effusion formation and cancer immune evasion.

predominance beyond the classic lymphocytic profile (Chae et al, 2021). Interestingly, pleural neutrophil deficiency (≤2500/mm³) was one of five parameters in the MAPED diagnostic score that predicted MPE with good accuracy (AUC ~ 0.82 derivation; ~0.72 validation), and importantly, correctly identified ~79% of cytology-negative MPE (Jia et al, 2024). These findings underscore the importance of excluding infection or adjusting for inflammatory confounders when interpreting neutrophil-based prognostic markers in MPE. Complementing these findings, Lim et al demonstrated in stage IV EGFR–wild-type NSCLC that a high pleural NLR was independently associated with early disease progression and shorter progression-free survival, even after adjustment for age, sex, infection status, tumor histology, and treatment, reinforcing the concept that local neutrophil predominance is a key adverse prognostic signal in MPE (Lim et al, 2020b). Beyond survival

prediction, serum NLR < 2.95 has also been identified as an independent protective factor against early symptomatic MPE recurrence following thoracentesis, with this association observed in both mutation-positive and -negative LUAD cohorts. It was further shown that in those patients with actionable mutations, the timely initiation of targeted therapy conferred additional protection against MPE recurrence (Xu et al, 2023). Lee et al further reported that a pleural fluid NLR ≥ 1.36 and a serum NLR ≥ 3.85 could also be used to individually identify patients with markedly shorter survival, and that combining these into a simple "smNLR" score (0, 1, 2) provided stepwise risk stratification (median OS 12.6, 4.4, and 1.6 months, respectively) independent of ECOG performance status and histology. Collectively, across multiple independent cohorts pleural neutrophil predominance, whether quantified morphometrically, as absolute proportion, or by pleural/serum NLR, has

emerged as a reproducible adverse prognostic marker in MPE, outperforming lymphocyte proportion and independent of systemic inflammation (Chan and Chan, 2024; Fonseka et al, 2022; Popowicz et al, 2021; Lim et al, 2020b; Nieto et al, 2019; Wu et al, 2019; Lee et al, 2017; Anevlavis et al, 2014).

Neutrophils exert many of their tumor-promoting effects via NETosis. These NETs are macroscopic fibrous, web-like structures composed of extracellular DNA, histones, and granule proteins that can trap pathogens, remodel tissue, and suppress immunity (Brinkmann et al, 2004). NETosis can be induced by tumor- and stroma-derived IL-8/CXCL8, CXCL1/CXCL2, G-CSF, TLR ligands, and platelet-derived HMGB1 and P-selectin, many of which are abundant in the pleural tumor microenvironment (Adrover et al, 2023). Once formed, NET-associated proteases (e.g., neutrophil elastase, MMP-9) and histone–DNA complexes can degrade extracellular matrix and basement membrane, citrullinate structural proteins via PAD4, increase vascular and mesothelial permeability, and release sequestered growth factors which facilitate pleural invasion, protein-rich capillary fluid leakage, and fibrinous septation (Albrengues et al, 2018; Cools-Lartigue et al, 2013).

In LUAD-derived MPE, although pleural neutrophil abundance and canonical surface phenotype were comparable to non-malignant effusions, the malignant pleural fluid milieu profoundly reprogrammed neutrophil function, enhancing viability and NETosis while attenuating oxidative burst (Mulet et al, 2022). This was accompanied by significantly elevated pleural levels of MPO, lactoferrin, MMP-9, and IL-8 compared to transudative controls, with each of these markers showing high diagnostic accuracy ($AUC \geq 0.78$) for distinguishing MPE. NETosis in response to malignant pleural fluid correlated with MMP-9, P-selectin, and soluble PD-L1 levels. Higher NETosis was independently associated with worse survival, whereas higher lactoferrin predicted improved survival. These findings position NET formation as a clinically relevant neutrophil effector program in MPE and identify soluble mediators within pleural fluid that may serve as biomarkers or therapeutic targets.

Mechanistically, NET proteases (e.g., NE, MMP-9) and histone–DNA complexes can directly increase vascular and mesothelial permeability, degrade extracellular matrix, and citrullinate structural proteins via PAD4, thereby facilitating pleural invasion, fluid leakage, and fibrinous septation (Demkow, 2021). Hypoxia, a consistent feature of the MPE microenvironment, further amplifies NETosis through HIF-1α-dependent pathways, sustaining a feed-forward inflammatory loop via the release of IL-8, CXCL1/2, G-CSF, and HMGB1, which recruits additional neutrophils and perpetuates effusion maintenance. The hypoxia- and glycolysis-enriched transcriptional milieu described in MPE immune cells (Huang et al, 2021b) may further potentiate ROS-dependent neutrophil effector functions and NETosis, providing a mechanistic bridge between mononuclear cell metabolic states and neutrophil-driven pathology in the pleural niche.

Consistent with broader cancer literature, NETs can carry PD-L1 on their DNA–protein complexes, directly suppressing cytotoxic T-cell function (Teijeira et al, 2020a). Host factors such as obesity and diabetes amplify NET release (Wong et al, 2015), suggesting potential modifiers of MPE risk and progression. In LUAD-MPE patients, the increased presence of NETs was correlated with poorer survival (Ito and Ogawa, 2022; Twaddell et al, 2021).

Therapeutically, approaches to modulate NETs in cancer include direct NET degradation (DNase I), inhibition of NETosis (PAD4 inhibitors, gasdermin-D blockade), and targeting upstream drivers such as IL-8/CXCR2, G-CSF, TLR pathways, and platelet–neutrophil interactions (Zhang et al, 2023d). In pleural disease, intrapleural coadministration (but not single-agent administration) of tissue plasminogen activator (t-PA) with DNase administration in patients with non-draining pleural infections is safe and effective (Rahman et al, 2011; Chong et al, 2021), suggesting a rationale to evaluate t-PA/DNase combinations in MPE regimens, particularly in patients with high NLRs or demonstrable NET enrichment on pleural fluid analysis (Fig. 3, pink).

### Inflammasome and DAMP–TLR signaling

Inflammasome activation requires a two-signal model: TLR4 signaling provides a priming signal (signal 1) via NF-κB-mediated upregulation of pro-IL-1β and NLRP3, while a second signal such as ATP, reactive oxygen species, or crystalline particles is necessary for full inflammasome assembly and activation (Mezzasoma et al, 2023; Sayan and Mossman, 2016; Karki and Kanneganti, 2019). In the pleural tumor microenvironment, inflammasome activity is further shaped by cell-type context: tumor cells, TAMs, CAFs, and MDSC-like populations each differentially contribute to priming versus activation, highlighting that pleural inflammasome engagement is compartmentalized across immune and stromal niches (Zhang et al, 2023c).

Notably, the DAMP protein high-mobility group Box 1 (HMGB1), an activator of TLR4 signaling, is secreted at high levels in mesothelioma and NSCLC. While it correlates with adverse prognosis in MPM and in molecularly defined NSCLC subsets, it is also elevated in benign effusions, limiting its diagnostic specificity (Suarez et al, 2023; Rrapaj et al, 2018; Jube et al, 2012; Takigami et al, 2024; Wu et al, 2020a). Importantly, mesothelioma studies have shown that HMGB1 is released in distinct isoforms: asbestos-induced necrosis predominantly yields non-acetylated

---

**Box 1   The clinical management of MPE**

1. Malignant pleural effusions (MPE) represent a disorder of lymphatic fluid production and/or drainage.
2. Accumulation and recurrence of MPE fluid in the pleural space causes a loss of chest wall and diaphragm compliance, leading to progressive shortness of breath.
3. Palliative management of shortness of breath due to MPE is performed by one of three clinical interventions: (1) Thoracentesis, (2) Tunneled pleural catheter insertion (TPC), or (3) Pleurodesis.
4. Serial thoracentesis for recurrent MPE is typically reserved for patients with survival <3 months, while TPC and pleurodesis are designed for longer-term management. (PMID: 22610520).
5. Resolution of MPE recurrence is independently associated with improved survival. (PMID: 3724080).
6. Depending on the drainage strategy, autopleurodesis has been shown to occur in 24–47% of patients managed by TPC (PMID: 27898215).
7. The combination use of TPC followed by chemical pleurodesis has been promoted as a method for achieving minimally invasive pleural symphysis (PMID: 29617585).

HMGB1, whereas tumor-intrinsic active secretion produces hyper-acetylated HMGB1, which constitutes the majority of circulating HMGB1 in MM and reflects active DAMP release into the pleural tumor milieu (Napolitano et al, 2016). In NSCLC, pleural-fluid analyses further demonstrate that HMGB1, along with IL-6 and IL-8, are significantly higher in MPE compared to benign effusions, while IL-1β remains selectively absent, supporting a model of a DAMP-primed but incompletely activated inflammasome milieu (Ma et al, 2023, 2017; Wu et al, 2020a).

In infectious effusions, HMGB1 strongly correlates with IL-1β and LDH as a triad of inflammasome–necrosis activity. By contrast, malignant effusions consistently show low IL-1β despite abundant HMGB1, pointing to functional suppression of inflammasome activation in MPE (Wu et al, 2020a; Ma et al, 2023, 2017). Metabolic cues may further explain this paradox: lactate can activate NLRP3 in macrophages via ROS, yet tumor-driven TGF-β–SMAD signaling induces autophagy that attenuates inflammasome responses, providing a mechanistic rationale for low IL-1β in MPE despite high DAMP burden (Zhang et al, 2023c).

Several studies have implicated HMGB1 in cancer pathogenesis, revealing that HMGB1 can exhibit both pro- and anti-tumorigenic effects, depending on its environmental context. Acute cellular release of HMGB1 activates immune clearance through recruitment and maturation of dendritic cells (DCs) and cytotoxic CD8+ T cells (CTLs) to the TIME; however, during sustained expression of HMGB1 in the TIME, immunosuppressive MDSCs and regulatory CD4+ T cells (Tregs) are recruited, dampening immune clearance of tumor cells, promoting drug resistance, and inhibiting cancer cell apoptosis (Livesey et al, 2012; Wild et al, 2012b; Huang et al, 2012; Kang et al, 2013; Liu et al, 2011). At the tissue level, HMGB1 has also been shown to translocate from the nucleus to the cytoplasm in NSCLC models and xenografts, a redistribution consistent with active release of HMGB1 as a functional DAMP into the tumor–pleural microenvironment (Ma et al, 2023). Within MPE, higher intrapleural HMGB1 has been associated with increased total CD45+ leukocyte burden but reduced CD14+ monocytes, reflecting a DAMP-conditioned remodeling of myeloid composition. Moreover, acellular MPE fluid potently suppresses monocyte chemotaxis and blunts LPS-induced TNF-α production, effects that are not reversed by HMGB1 neutralization, suggesting that multiple soluble mediators beyond HMGB1 contribute to impaired myeloid effector function in the pleural niche. In addition, higher HMGB1 levels correlated with reduced γδ TCR diversity, while MPE fluid promoted γδ T-cell proliferation yet suppressed TNFα and IL-10 output, indicating that HMGB1-rich pleural environments may foster expansion but functional restraint of innate-like T-cell subsets (Soloff et al, 2020). Recent pleural-fluid analyses further support a context-dependent role for HMGB1 in MPE biology: in NSCLC patients with EGFR/ALK driver mutations, elevated HMGB1 in effusion fluid correlated negatively with overall survival, underscoring its prognostic significance in molecularly defined subsets (Takigami et al, 2024).

Beyond HMGB1, inflammasome pathways themselves are emerging as pro-tumorigenic mediators in pleural disease. NLRP3 has emerged as the central inflammasome sensor mediating asbestos-induced mesothelial injury and mesothelioma pathogenesis (Thompson et al, 2017; Mossman et al, 2013). Complementary work highlights other regulators of pleural inflammasome biology:

the stress-response mediator HO-1, which was selectively elevated in infectious effusions, may limit HMGB1 release and inflammasome-driven injury, though its significance in MPE remains unexplored. ATP metabolism also plays a role: blockade of CD39-mediated ATP hydrolysis enhances NLRP3–IL-18 signaling and expands CD4+ and CD8+ effectors, suggesting that evaluating ATP/CD39/CD73 pathways in pleural effusions may serve as effusion biomarkers of inflammasome activity (Zhang et al, 2023c).

Finally, while TLR4 is often regarded as a driver of pro-tumor inflammation, recent work suggests it may actually inhibit MPE formation by modulating Th1/Th17 responses, highlighting the context-dependent roles of DAMP–TLR signaling in the pleural tumor microenvironment (Xu et al, 2015). In pleural malignancy, transcriptomic analyses now show that DAMP and sensing-receptor programs delineate two molecular subtypes: Inflammatory DAMPs (inflamed TIME, better overall survival) versus Nuclear DAMPs (immune-suppressed, worse survival), supporting a model in which DAMP–PRR–inflammasome signaling helps set pleural immune tone (Liu et al, 2023b). Key discriminative markers include canonical DAMPs (HMGB1, CALR, HSPs), their receptors (TLR2/3/4/7, AGER/RAGE), and inflammasome sensors (NLRP3, AIM2), linking ICD-associated pathways to pleural tumor biology (Liu et al, 2023b). The Inflammatory DAMPs subtype also showed higher CD8A, PD-1/PD-L1, and CTLA-4 expression, as well as enriched ICI responses, highlighting the clinical relevance of DAMP–PRR activity to MPE-related immunobiology. Notably, Takigami et al also introduced a whole-cell pleural fluid assay in which anti-PD-1 stimulation elicited CD8+ T-cell IFN-γ release, demonstrating the feasibility of functional MPE assays that could be extended to future inflammasome readouts such as IL-1β or IL-18 secretion. Additional evidence suggests that therapy itself may feed into pleural inflammasome signaling: DNA-damage DAMPs released after radiotherapy activate AIM2, NLRP3, and cGAS–caspase-11 pathways, while targeted inhibitors such as selumetinib or BRAF inhibitors can provoke macrophage or stromal inflammasome activation, linking treatment-induced stressors to pleural inflammasome engagement (Zhang et al, 2023c).

Other studies have also provided supportive evidence for investigating inflammasomes in MPE. For example, complement protein C5a has been identified to trigger inflammasome-mediated secretion of IL-1β from monocytes, and promote increased homing of intermediate monocyte subtypes (CD14++CD16+) to the pleural space by causing increased expression of the pleural mesothelial cell-derived chemoattractant CCL2 (commonly called monocyte chemoattractant protein 1, MCP-1) (Luo et al, 2022b). As such, possible inflammasome-mediated roles in MPE development and sustenance of MPE should be considered for further study (Fig. 3, green).

### Dendritic cells

Dendritic cells (DCs) are present in MPE, but are functionally polarized toward an immature immunoregulatory state that contributes to T-cell dysfunction and local immunosuppression. Single-cell RNA sequencing and flow cytometry studies have identified both conventional DC subsets (cDC1 and cDC2), inflammatory DCs (infDCs), and plasmacytoid DCs (pDCs) in

pleural fluid from LUAD patients, with transcriptional evidence of impaired antigen processing and presentation pathways compared to DCs from peripheral blood or tumor tissue (Huang et al, 2021b). InfDCs are numerically minor (~1% of light-density cells) and express CD206, CD14, and CD11b, remaining functionally immature unless stimulated by TLR agonists (Gu et al, 2020). pDCs in MPE exhibit a dysfunctional, pro-tumorigenic phenotype, with high expression of gene sets associated with poor prognosis in lung cancer (Wu et al, 2023b). Pleural DCs often display reduced expression of co-stimulatory molecules and increased expression of inhibitory ligands, reinforcing this suppressive state (Ge et al, 2024). The relative scarcity of cDC1s, critical for cross-priming of cytotoxic T cells, may limit anti-tumor immunity in MPE. In addition, the pleural microenvironment, characterized by hypoxia and glycolytic reprogramming, further impairs DC maturation and antigen-presenting capacity, limiting effective T-cell priming (Huang et al, 2021b). Senescent tumor cells expressing HLA-E disrupt the NK cell–DC–T cell axis by suppressing NK-derived XCL2, thereby limiting the recruitment of cross-priming cDC1s and driving the exhaustion of PD-L1$^+$ DC2-like subsets. In parallel, pleural mediators such as CXCL16, BAG6, and IL-7 are associated with poor prognosis (Tsai et al, 2025). Murine studies further show that the CD93–CCL21–CCR7 axis governs pleural DC recruitment, with anti-CD93 therapy enhancing DC influx and tumor control in a DC- and CCR7-dependent manner; in patients, serum EV-miR-5193 and C1q levels correlate with checkpoint inhibitor sensitivity, supporting the translational relevance of this pathway (Zhang et al, 2024a). Intrapleural delivery of LNP-CDN, while preferentially taken up by TAMs, also activates pleural DCs, inducing type I IFN and antigen-presentation programs across both effusion and pleural tumor compartments; ex vivo exposure of human NSCLC MPE similarly reprograms pleural DCs, underscoring their therapeutic tractability and contribution to checkpoint synergy (Liu et al, 2022). Together, these data highlight that although pleural DCs are numerically scarce and functionally impaired, they remain pharmacologically reprogrammable in situ.

In addition to being functionally deficient, DCs can also act as active drivers of MPE pathogenesis. Murine studies demonstrate that DCs can act as direct effectors of MPE progression: in an intrapleural Lewis lung carcinoma model, IL-10 signaling in pleural DCs activates STAT3 and induces secretion of tissue inhibitor of metalloproteinases 1 (TIMP1), with DC-specific Il10ra deletion markedly reducing MPE burden (Zhao et al, 2021). Complementary analyses of human monocyte-derived DC datasets corroborate this IL-10-responsive TIMP1 program, underscoring the translational relevance of DC-intrinsic effector functions in the pleural niche. As a result, pleural T cells display upregulated inhibitory receptors and diminished effector function (Donnenberg et al, 2023; Dhupar et al, 2020). Given their ability to bridge innate and adaptive immunity, therapeutic strategies that enhance DC recruitment, maturation, and antigen presentation in the pleural space could augment the efficacy of immunotherapy. Promising avenues include: (i) innate agonists such as TLR ligands and STING-activating nanotherapies that recondition DCs (Liu et al, 2022); (ii) metabolic modulators and intrapleural checkpoint blockade combinations (Plesca et al, 2022; Cheng et al, 2025); and (iii) cellular therapies using ex vivo-expanded DCs or T cells, which will likely require pleural microenvironmental conditioning for durable responses (Principe et al, 2021).

## Natural killer (NK) cells/innate lymphoid cells (ILCs)

Natural killer (NK) cells in MPE are present but at relatively low frequency compared to T cells and macrophages, particularly when contrasted with peripheral blood, where cytotoxic CD56$^{dim}$ CD16$^+$ NK subsets are markedly more abundant and are especially depleted in MPE (Marcq et al, 2017). They exhibit profound functional impairment, including reduced perforin content, potentially reinforced by pleural exosomal miRNAs such as miR-3120-5p, which bioinformatically target PRF1, and increased expression of inhibitory receptors such as CD94/NKG2A and PD-1, which directly correlates with diminished cytotoxicity against tumor cells (Zhang et al, 2023b; Pace et al, 2011). NKG2A is upregulated on both NK and cytotoxic T cells, while HLA-E expression on senescent tumor cells further drives exhaustion. Phenotypic skewing toward a decidual-like, pro-angiogenic profile, marked by enrichment of CD56$^{bright}$ CD16$^-$ subsets with CD49a$^{hi}$ CD69$^{hi}$ CD57$^{low}$ expression, has also been observed, particularly in tumor-associated effusions, with these NK cells producing VEGF and PlGF that promote neovascularization in the pleural niche (Bosi et al, 2018). Hypoxia and glycolytic reprogramming within the pleural space further suppress NK cell effector functions by disrupting mitochondrial integrity and cytotoxic granule release, while also impairing NK cell trafficking and retention in the pleura (Tumino et al, 2023; Wu et al, 2025a). Pathway enrichment of pleural exosomal miRNAs further implicates FcγR-mediated phagocytosis and Rap1 signaling, suggesting EV-driven modulation of NK/ILC synapse formation and trafficking (Zhang et al, 2023b). Prognostically, higher PF/PB ratios of cytotoxic CD16$^+$ and CD57$^+$ NK cells have been linked with improved survival outcomes (Lara et al, 2019). Nevertheless, NK cells retain the capacity for functional rescue: IL-15 stimulation restores granzyme B expression and cytotoxic activity, thereby overcoming pleural fluid-mediated suppression (Croxatto et al, 2017). Exogenous IL-2 shows only transient benefit in MPE settings and is further suppressed by TGF-β or soluble pleural factors, limiting its efficacy (Sivori et al, 2021; Bosi et al, 2018). IL-15 is superior to IL-2 in this context, being less susceptible to TGF-β-mediated inhibition, preferentially avoiding regulatory T-cell expansion, and demonstrating a more favorable safety profile with lower risk of capillary leak syndrome (Yang and Lundqvist, 2020). Building on this, adoptive NK cell therapy using IL-15-primed NK cells isolated from pleural fluid has emerged as a promising therapeutic avenue. Beyond cytokine priming, antigen-specific Fc engagement offers another path to functional rescue: in orthotopic mesothelioma models, anti-podoplanin antibodies (notably NZ-12) elicited robust NK-mediated ADCC, reducing both tumor burden and effusion volume; pemetrexed further enhanced efficacy by upregulating podoplanin expression on tumor cells (Abe et al, 2016). The NKG2A:HLA-E axis is a key checkpoint in MPE, and blockade of NKG2A may enhance NK cell antitumor activity (Giotti et al, 2024).

Innate lymphoid cells (ILCs) beyond NK cells, including ILC1, ILC2, and ILC3, are present at low frequency in MPE (Tumino et al, 2019). In a 54-patient cohort comprising mesothelioma and LUAD, ILC3s emerged as the predominant subset and retained canonical cytokine functionality, with ILC1 producing IFN-γ, ILC2 secreting IL-5 and IL-13, and ILC3 generating IL-22/IL-17 upon stimulation. ILC3s are the predominant non-NK ILC subset and express functional PD-1. Paired PD-L1 expression is demonstrated on tumor cells and pleural tumor-derived cell lines,

underscoring inhibitory PD-1:PD-L1 crosstalk in the pleural niche. While ILC2-mediated type 2 cytokine production could theoretically contribute to fibrosis and pleural remodeling, this remains speculative in the malignant setting (Joseph et al, 2025; Otaki et al, 2023; Bruchard and Ghiringhelli, 2019). ILC3-like cells may participate in the formation of tertiary lymphoid structures in other cancers, but their contribution to MPE immune architecture remains unconfirmed (Bennstein and Uhrberg, 2022). Notably, pleural immune cell transcriptomes exhibit an enrichment of hypoxia- and glycolysis-related programs, which could further dampen NK and ILC effector functions (Retamal et al, 2025; Borde and Matosevic, 2023; Huang et al, 2021b).

### Monocytes/MDSC-like cells

Monocytes and MDSC-like populations are prominent in the MPE immune landscape. Single-cell profiling has revealed inflammatory monocyte clusters with high S100A8/9 expression and immunosuppressive transcriptional signatures, accompanied by ligand–receptor analyses implicating tumor- and mesothelial-derived CCL2 and CCL7 as dominant CCR2-dependent drivers of monocyte recruitment (Huang et al, 2021b; Wu et al, 2023b). Consistent with these transcriptomic programs, functional studies in human pleural samples show that acellular MPE fluid suppresses monocyte chemotaxis and blunts cytokine production, reinforcing their impaired effector capacity (Soloff et al, 2020). In addition to transcriptional dysfunction, comparative analyses demonstrate that MPE are enriched for intermediate $CD14^{++}CD16^{+}$ monocytes, whereas nonclassical $CD14^{+}CD16^{++}$ monocytes predominate in tuberculous effusions, reflecting differential chemokine dependence (CCR2–CCL2 vs. CX3CL1–CX3CR1) and highlighting subset composition as a potential diagnostic marker (Luo et al, 2022a). Complement-driven C5a–C5aR1 signaling further amplifies this process by inducing IL-1β secretion and enhancing CCL2-mediated recruitment of intermediate monocytes to the pleural space (Luo et al, 2022b). In LUAD-derived MPE, these cells often coexist with monocyte-derived macrophages exhibiting tumor-promoting phenotypes (Bruschini et al, 2022). Clinically, elevated pleural CCL2 (also known as monocyte chemoattractant protein-1, MCP-1) has been correlated with increased effusion volume, reduced $CD8^{+}$ T-cell infiltration, and shorter survival, underscoring the prognostic relevance of the CCL2–CCR2 axis in MPE (Tekin et al, 2025)

While the expansion and suppressive activity of MDSCs are well-established across cancer types (Lim et al, 2020a), recent mechanistic analyses have refined this model by highlighting the STAT3-centered regulatory network, S100A8/A9-driven feedback signaling, and metabolic rewiring through arginase-1, inducible nitric oxide synthase, and reactive oxygen species as convergent pathways sustaining myeloid immunosuppression in the pleural niche (Santibanez, 2025; Veglia et al, 2021). Within MPE, these suppressive myeloid programs are further sustained by local CCL2 and CXCL8 gradients and oxidative metabolic constraints, although direct functional tracing in the pleural compartment remains limited (Ge et al, 2024). These cells can inhibit T cell proliferation, produce reactive oxygen species, and metabolically reprogram the microenvironment through the activity of arginase-1 and inducible nitric oxide synthase. Nanotherapeutic reprogramming of MPE myeloid cells has been shown to reduce immunosuppression and enhance anti-tumor responses in preclinical models (Song et al,

2021), underscoring the translational potential of targeting this compartment.

### Eosinophils, basophils, and complement

Eosinophils and basophils are rare in MPE, and their functional impact remains poorly characterized. Recent work has shown that CCL11/CCR3-dependent eosinophil recruitment alleviates malignant pleural effusions. CCR3 blockade reduces pleural eosinophils and worsens disease, while recombinant CCL11 or eosinophil augmentation increases eosinophilia and improves outcomes (Zhang et al, 2024c; Matos et al, 2014). In MPE mouse models, tumor-derived CCL11 expression was sufficient to drive eosinophil recruitment to the pleural space and restrain effusion accumulation. These findings identify a mechanism of immune modulation in MPE, where eosinophil recruitment via the CCL11/CCR3 axis opposes the immunosuppressive environment typically associated with MPE, suggesting dual roles for CCL11-driven cytokines in regulating effusion dynamics and immune tone. Conceptually, type 2 immune responses, mediated by ILC2s, Th2 cells, and eosinophils, have been linked to fibrotic pleural remodeling in non-malignant contexts (Kwon et al, 2013), but evidence for such activity in LUAD-associated MPE is minimal. Basophil involvement is largely speculative, as these cells are infrequently detected in single-cell datasets from pleural fluid (Huang et al, 2021b). The complement system, in contrast, appears more consistently active in MPE. Complement components, such as C3 and C5a, can modulate tumor-associated myeloid cell recruitment and suppress anti-tumor immunity (Magrini et al, 2022; Afshar-Kharghan, 2017). While the membrane attack complex (MAC) has been reported to deposit on tumor cells in certain solid tumors, its presence and functional significance in LUAD-MPE have not been systematically examined. Given the complement's dual potential for tumoricidal and tumor-promoting effects, dissecting its spatial and functional activity in pleural metastases may reveal novel immunomodulatory targets.

## Adaptive immune cells in MPE

### CD4+ T cell subsets have competing functions in MPE

Malignant pleural effusions are defined clinically by a high frequency of lymphocytes (Yam, 1967), with the highest frequency subpopulation comprising helper T cells (CD4+ T cells) (Yi et al, 2021). Multiple effector helper T cell subsets exist in MPE. Th1 cells are frequently observed, as roughly 45% of CD4+ T cells from MPE can produce interferon-γ (IFN-γ), the characteristic cytokine produced by Th1 cells (Nieto et al, 2019). Th1-derived IFN-γ is crucial for activating cytotoxic T cells and macrophages, suggesting potential local anti-tumor activity and correlating with improved chemotherapy responses and survival in mesothelioma and lung adenocarcinoma patients (Klotz et al, 2024). In contrast, canonical Th2 cells (producers of IL-4, IL-5, IL-13) appear less prominent and are thought to contribute minimally to MPE pathogenesis. Th17 cells (producers of IL-17) are also present at higher frequencies in MPE than in peripheral blood, and their enrichment is associated with elevated IL-6 and IL-23 signaling within the pleural space (Niu and Zhou, 2022). Recent data further identify IL-26 as a pleural-enriched cytokine produced mainly by Th17 and $CD4^{+}IL\text{-}22^{+}$ (Th22) cells, which acts autocrinely to promote Th22

differentiation and expansion through STAT3 activation (Niu et al, 2021). IL-26 exposure increases Ki-67 expression and IL-22 secretion in CD4$^+$ T cells, linking IL-6/IL-23/IL-23-driven pleural inflammation to CD4$^+$ subset plasticity. High pleural IL-26 levels correlate with Th22 frequency and predict worse overall survival, highlighting a distinct IL-26–Th22 axis in the pathogenesis of MPE. Notably, bulk flow-cytometric analyses of human exudative, lymphocyte-dominant pleural effusions show that total CD4$^+$ and CD8$^+$ frequencies do not significantly differ between malignant and tuberculous etiologies, reinforcing that functional subset composition rather than absolute CD4$^+$ abundance defines the malignant pleural immune landscape (Mehraban et al, 2022).

Moreover, malignant pleural effusions are enriched for CD69$^+$/ CD103$^+$ CD4$^+$ tissue-resident memory (Trm) cells that exhibit diminished IFN-γ and TNF-α production, reduced granzyme B and perforin expression, and elevated PD-1 levels relative to matched blood, indicating compartment-specific functional exhaustion within an IL-6-rich pleural milieu (Mao et al, 2023). However, recent flow-cytometric and single-cell analyses of human NSCLC-associated MPE quantify a distinct CD4$^+$ Trm subset (~6% of CD4$^+$ T cells) expressing canonical residency genes (ITGAE/CD103, ITGA1/CD49A, CXCR6, RGS1) and low CCR7 and IL7RA (Tilsed et al, 2025). Despite co-expression of PD-1, TIGIT, and CD39, these pleural Trm retain robust IFN-γ, TNF-α, and CD107a responses upon stimulation, indicating that checkpoint positivity does not uniformly signify dysfunction in this context. CD69, but not CD103, correlates most strongly with effector cytokine production, suggesting that pleural Trm function is governed by local residency signaling rather than classical exhaustion. Single-cell transcriptomic profiling of LUAD-associated MPE has similarly revealed CD4$^+$ clusters co-expressing PD-1, CTLA-4, and TIGIT, with reduced IFN-γ and GZMB expression, consistent with an exhausted helper T-cell phenotype under metabolic constraint. Ligand–receptor network analyses further highlighted MHC-II–CD74 and TGF-β–TGFBR1 signaling between epithelial and macrophage compartments and CD4$^+$ T cells, indicating tumor-driven pathways of CD4$^+$ tolerization within the pleural microenvironment (Huang et al, 2021b).

In addition, experimental models demonstrate that pleural IL-10 acts as a critical immunosuppressive axis limiting local Th1 immunity and promoting effusion formation. Loss of IL-10 signaling reduces vascular permeability and tumor burden, whereas IL-10 exposure suppresses IFN-γ production by CD4$^+$ cells through a microRNA-dependent mechanism (Zhai et al, 2020) Specifically, IL-10 upregulates miR-7116-5p, which targets the G-protein-coupled receptor GPR55 and attenuates ERK-mediated Th1 signaling, providing a post-transcriptional pathway linking anti-inflammatory cytokine signaling to suppressed CD4$^+$ effector function within MPE. These findings extend human observations of pleural T-cell exhaustion by identifying an IL-10/miRNA/ERK axis that actively represses Th1 responses and facilitates tumor persistence in the pleural cavity.

Single-cell transcriptomic comparisons of paired primary lung tumors and malignant pleural effusions from patients treated with first-line immunochemotherapy further reveal that CD4$^+$ T cells remain the dominant T-cell subset in MPE but undergo post-immunotherapy transcriptional reprogramming characterized by enrichment of naive-like and metabolically active states. Pleural CD4$^+$ T cells upregulate oxidative phosphorylation, DNA repair, and MYC target pathways alongside broad IRF9 activation, reflecting interferon-associated and metabolic adaptation within the effusion environment (Wu et al, 2025b). In parallel, microRNA-driven regulation further shapes CD4$^+$ fate within MPE: elevated miR-16-5p expression in NSCLC-associated effusions suppresses IFN-γ-regulated differentiation of naive CD4$^+$ cells into CD69$^+$ memory T-helper phenotypes, thereby attenuating pleural Th1-like memory generation and potentially contributing to impaired immunotherapeutic responsiveness (Sun et al, 2024). This finding adds a post-transcriptional dimension to CD4$^+$ subset remodeling, complementing cytokine-driven and metabolic mechanisms described above.

### Regulatory T cells mediate tumor-promoting immunosuppression in MPE

Especially true for lymphocytic MPE, the poor clearance of malignant cells suggests that increased suppressive T-cell activity and/or impaired cytotoxic T-cell functionality are key driving factors in MPE formation. Several molecules have been implicated in MPE T cell suppression, including TAM-produced TGF-β and HMGB1 (Li et al, 2016). HMGB1 was shown to enhance the proliferation of γδT cells in the context of MPE. While γδT cells can exhibit both anti- and pro-tumorigenic effects, when cultured in acellular MPE fluid, γδT cells were able to expand. However, they were impaired in their ability to produce anti-tumorigenic cytokines, resulting in terminal differentiation and γδT cell exhaustion. As previously discussed, HMGB1 also recruits Tregs, resulting in the secretion of the immunosuppressive cytokine IL-10 by these cells.

Additionally, several subtypes of Tregs, including FoxA1$^+$ and FoxP3$^+$ Tregs, have been shown to be present at significantly higher frequencies in MPE compared to benign effusions (Budna et al, 2018; Liang et al, 2019). Evidence suggests that a predominant subtype of Tregs in MPE (FoxP3$^+$ Tregs) expresses immunosuppressive markers, such as CTLA-4 and CD25, at higher frequencies compared to otherwise benign pleural fluid (pleural effusions driven by osmotic or oncotic abnormalities, such as congestive heart or liver failure). This may indicate a more prominent immunosuppressive response in the context of malignancy, potentially playing a role in differences between the body's ability to resolve benign pleural effusions compared to MPE (Wang et al, 2023; Tano et al, 2023; Niu and Zhou, 2022; Yi et al, 2021; Ye et al, 2012; Chen et al, 2005).

Interestingly, the proportion of Th17 cells in MPE inversely correlates with the abundance of Tregs (FOXP3 + CD4 +), implying that a Treg-rich environment may actively restrain Th17 differentiation in the effusion (Principe et al, 2021). Within MPE, Tregs emerge as key mediators of immunosuppression, suppressing effector T cell responses via cell-contact and inhibitory cytokines (IL-10, TGF-β), effectively dampening anti-tumor cytotoxicity and associating with shorter overall survival (Yang et al, 2015a). In particular, *maximally* suppressive Tregs characterized by tumor necrosis factor receptor type II (TNFR2) expression are enriched in MPE, and their abundance correlates with greater tumor cell counts, larger effusion volumes, and worse patient outcomes (Ye et al, 2020; Cheng et al, 2025). Mechanistically, lactate accumulation within the pleural fluid promotes histone

lactylation and upregulation of TNFR2 on Tregs, enhancing their suppressive phenotype and reinforcing local tolerance (Xue et al, 2024). The tumor microenvironment actively drives Treg recruitment; for example, CCL22 levels are significantly elevated in MPE (especially in cases of lung cancer) and are primarily produced by tumor-associated macrophages in the pleura. High CCL22 levels are associated with poorer survival (Wang et al, 2019a). There is evidence that pro-inflammatory CD4 + T cells can paradoxically amplify immunosuppression: Th1 and Th17 cells in MPE produce TNF-α, which signals via TNFR2 on Tregs to increase further Treg suppressive activity (upregulating CTLA-4 and PD-L1 on these Tregs) (Bruschini et al, 2022). Thus, even the ostensibly anti-tumor Th1/Th17 response can unintentionally boost Treg-mediated tolerance, highlighting the complex feedback loops in this environment. Recent preclinical work suggests that targeting these feedback circuits, for example, through dual IL-6/PD-L1 blockade, can reprogram CD4$^+$ T cell polarization and remodel the tumor microenvironment toward a more immunostimulatory state (Cheng et al, 2025). In parallel, novel bioengineered immunotherapies show potential to restore Th1-driven immunity and counteract Treg dominance in MPE (Chen et al, 2025b).

### Cytotoxic T cells (CTLs) have impaired anti-tumor functionality in MPE

Although some MPE resident CTLs can execute tumor-reactive cytotoxic functions (as evidenced by their ability to lyse autologous tumor cells ex vivo), most patients have MPE-derived CTLs with impaired functionality that do not display these tumor-reactive capabilities. Although patients with lymphocytic MPE, low NLRs, and "hot" immune landscapes have better outcomes than patients with neutrophilic MPE, high NLRs, and "cold" immune landscapes, the inability of T cells to execute cytotoxic functions may provide insight into why patients with lymphocyte-predominant MPE still have poor outcomes. Evaluation of CTLs from patient MPE samples consistently shows that these populations are functionally suppressed within the MPE TIME. Mediators of this cytotoxic T cell suppression include HMGB1, which has been shown to inhibit CTL effector function, resulting in decreased tumor-associated cytotoxicity (Fig. 3, orange) (Wild et al, 2012a). Studies have also demonstrated impairments in the effector functions of MPE resident CTLs when they are cocultured with MPE-derived MDSCs, TAMs, Tregs, and acellular components of MPE. This was evidenced by decreased CTL cytotoxicity and reduced ability to produce interferon (IFN-γ), even in the presence of the CTL stimulant IL-2. When CTLs were cultured in a 1:1 ratio of traditional culture medium to acellular MPE fluid, CTL effector capabilities were significantly decreased as compared to CTLs cultured in traditional medium alone. Lactate and lactate dehydrogenase, which are known to be highly present in MPE fluid, were also shown to correlate with CTL dysfunction. This may, in part, explain the challenges in establishing T cell therapies for MPE, as attempts to expand MPE-derived CTLs using the same methodology as adoptive cell transfer ex vivo result in CTLs becoming exhausted and ineffective. However, recent single-cell RNA-seq profiling of matched lung tumor biopsies versus MPE has demonstrated that MPE have a higher proportion of naive T cells and a lower proportion of exhausted T cells and Tregs, suggesting a potential distinct opportunity for immunotherapy interventions (Mahmood et al, 2024).

### B cells in malignant pleural effusions

In addition to T cells, B cells are also present in MPE, where they can function as antigen-presenting cells, secrete immunoglobulins, and modulate T cell activity (Huang et al, 2021b). Single-cell mapping from the same cohort further delineated seven B-cell and four plasma-cell subsets, with CD24$^+$CD27$^+$ regulatory B cells enriched in pleural fluid and plasma cells predominating in blood. Pleural B cells exhibited hypoxia-linked glycolytic and interferon-responsive transcriptional programs, characterized by upregulated STAT1, IRF9, PKM, and ENO1 expression, consistent with the global metabolic rewiring described in the pleural immune compartment. Moreover, pleural Regulatory B cells (Bregs, typically CD24$^{Hi}$CD27$^+$) expressed BAFF-R (TNFRSF13B) and LGALS1 (Galectin-1) and displayed preferential predicted interactions with CD4$^+$ T-cell subsets over CD8$^+$ T cells, supporting an immunoregulatory rather than cytotoxic-supportive role within the pleural niche. Recent pan-cancer single-cell mapping has further expanded the taxonomy of tumor-associated B cells, identifying stress-response memory B cells and tumor-associated atypical B cells (TAABs) as phenotypically distinct and functionally relevant intratumoral states characterized by clonal proliferation, IgG-skewed antibody-secreting activity, and preferential engagement with activated CD4$^+$ T cells (Yang et al, 2024). These archetypes provide a conceptual framework for interpreting pleural B-cell heterogeneity and raise the possibility that TAAB-like programs, rather than classical plasma or Breg states, may underlie adaptive B-cell remodeling within the MPE microenvironment. Notably, B cells in MPE often exhibit an activated phenotype with upregulated MHC-II, CD80/86, and PD-L1 expression, which can skew CD4$^+$ T helper responses toward Th2-polarized inflammation and promote effusion formation, as demonstrated in murine models (Wu et al, 2018). Complementing these findings, flow-cytometric analyses of LUAD-associated MPE have shown enrichment of CD19$^+$CD24$^{hi}$CD27$^+$ Bregs and loss of CD19$^+$CD27$^-$IgD$^+$ naive B cells, accompanied by elevated CD1C, CD40, CD80, and CD86 expression and production of IL-10, TGF-β, and TNF-α (Shao et al, 2023b). In a more extensive cohort analysis of 143 LUAD MPE and matched blood samples, CD24$^+$CD27$^+$ B cells were again confirmed as the dominant pleural subset and displayed a transcriptionally and phenotypically distinct regulatory signature characterized by high IL-10 and CD39 expression together with downregulation of CD5, CD25, CD38, CD71, PD-1, and PD-L1 (Liang et al, 2025). This checkpoint-independent regulatory state contrasted with CD27$^+$CD38$^+$ B cells, which retained higher PD-1/PD-L1 expression but were reduced in frequency in LUAD MPE, suggesting a subset-specific divergence between immunoregulatory and plasma precursor programs. Correlative analyses revealed that pleural CD24$^+$CD27$^+$ frequencies associated positively with eosinophil and total WBC counts, implicating potential B-eosinophil cross-talk in shaping local immune balance. These findings establish CD24$^+$CD27$^+$ Bregs as a defining feature of LUAD MPE with diagnostic and prognostic potential, reinforcing IL-10$^+$ B cells as key regulators of pleural immune tone.

These pleural Bregs suppress Th1 and promote Treg differentiation in vitro, consistent with their immunosuppressive role. In parallel, Th17-associated cytokines IL-17 and IL-21, which are elevated in MPE (Niu and Zhou, 2022), can further influence pleural B-cell dynamics by inducing BAFF/TNFSF13B production from mesothelial and myeloid cells, as well as by directly promoting

B-cell activation, proliferation, and class-switch differentiation. This Th17-BAFF/IL-21 axis likely reinforces B-cell persistence and differentiation within the inflamed pleural niche, complementing BAFF-dependent signaling networks identified in single-cell studies. Single-cell transcriptomic comparison of primary LUAD tumors and MPE further demonstrated that pleural B cells are skewed toward a naive-enriched and plasma-cell-depleted composition, accompanied by heightened IRF9 activity and trend-level enrichment of Type I interferon-responsive pathways (Wu et al, 2025b). This interferon-associated transcriptional program parallels the IRF9-driven metabolic adaptation previously observed in pleural CD4$^+$ T cells, suggesting a shared IFN-I-tuned state across adaptive lymphocytes within the effusion microenvironment.

Beyond their antigen-presenting capacity, recent single-cell analyses have identified heterogeneous B-cell subsets within MPE, including activated, regulatory, and plasma-differentiated populations, with distinct transcriptional programs (Yang et al, 2024). When contextualized with the emerging taxonomy of tumor-associated B cells, these pleural subsets likely occupy a narrower functional spectrum dominated by regulatory and naive phenotypes rather than proliferative or antibody-secreting TAAB-like populations, aligning with the largely non-clonal BCR architecture reported in MPE (Huang et al, 2021b; Wu et al, 2025b). Transcriptomic profiling of pleural Bregs versus naive B cells further revealed alternative-splicing and immune-inflammatory pathway enrichment, including a CD27-AS1 isoform shift associated with high CD27 expression (Shao et al, 2023b). In addition, network analyses from Wu et al highlight enhanced B-cell–myeloid crosstalk within MPE, characterized by upregulated BAFF/ TNFSF13B, CD22, IL16, and SELPLG signaling axes that may sustain B-cell activation and survival in the pleural niche (Wu et al, 2025b). Bregs are enriched in the pleural fluid relative to peripheral blood, particularly in lung adenocarcinoma-associated MPE, suggesting a role in suppressing effective anti-tumor immunity through IL-10 and CD39-mediated immunoregulation (Liang et al, 2025). IL-10$^+$ Bregs suppress anti-tumor immunity by inhibiting effector T-cell responses and fostering a tolerogenic microenvironment (Horii and Matsushita, 2021; Cerqueira et al, 2019). Consistent with these findings, IL-10 produced by B cells and other immune populations directly promotes MPE formation in preclinical models by dampening Th1 responses and facilitating tumor growth and vascular leakage (Zhai et al, 2020). Human studies further confirm elevated IL-10 concentrations in MPE fluid, which are associated with local immunosuppression and reduced cellular immunity (Yanagawa et al, 1999; Chen et al, 1996). Collectively, these data support a dual role for pleural B cells as antigen-presenting and immunoregulatory effectors, whose transcriptional plasticity and splicing adaptations may shape the immune microenvironment of MPE.

In LUAD MPE, an "immune hot" environment characterized by an abundance of B and T cells correlates with improved patient survival (Wu et al, 2022). Patients with a high B-cell-to-leukocyte ratio and low neutrophil counts in pleural fluid have better outcomes, highlighting the prognostic value of adaptive immune infiltration (Wu et al, 2019). Integration of B-cell infiltration scores with alternative-splicing signatures in tumor datasets suggests that B-cell abundance is most strongly linked to favorable prognosis in metastatic LUAD, though not all B-cell subsets confer equal benefit (Shao et al, 2023b). Notably, the naive-skewed, IRF9-high B-cell

state identified by Wu et al may represent a context-dependent adaptive response to the pleural interferon-rich microenvironment, the functional consequences of which remain to be defined (Wu et al, 2025b). In breast cancer MPE, single-cell analysis has confirmed the presence of B cells (albeit at low frequencies within the effusions), although their function and prognostic impact remain unknown (Whitfield et al, 2023).

## Non-cellular drivers of MPE

Appropriate lymphatic circulation is a crucial factor in maintaining fluid homeostasis within the pleural space (Marazioti et al, 2014). In healthy individuals, pleural fluid circulates through the lymphatic vasculature in the pleural lining around the lungs and the diaphragm. There is a fine-tuned balance of pleural fluid production and reabsorption within the pleural space that maintains homeostasis of respiratory physiology (Solari et al, 2022; Zocchi, 2002). In the case of lymphatic obstruction caused by tumor metastases invading the pleural-associated lymphatic vessels, physical disruption of lymphatic drainage results in fluid over-accumulation and the formation of MPE (Bielsa et al, 2008a; Feller-Kopman and Light, 2018).

In addition to obstructed lymphatic drainage, multiple studies have demonstrated increased vascular permeability in the context of MPE. This characteristic is used clinically as diagnostic criteria, Light's Criteria, to differentiate between transudative and exudative fluid (Porcel and Light, 2006). Unlike transudates, which occur when fluid transits out of the vasculature due to disruptions in osmotic and/or oncotic pressure, exudates result from increased vascular permeability caused by inflammation and are one of the defining characteristics of MPE (Wilcox et al, 2014). Experimentally, this can be demonstrated through visualization of Evans' blue dye in the pleural cavities of mice with MPE as compared to the pleural cavities of healthy mice (Zebrowski et al, 1999; Yano et al, 2000a, 2000b). In addition to lymphatic obstructions that modulate the rate of pleural fluid egress, increased vascular permeability results in a rate of fluid ingress to the pleural space at a rate faster than it can be filtered out and reabsorbed, causing a constant loop of effusion fluid continuously entering the pleural space, where potentially obstructed lymphatics are unable to drain it adequately (Gonnelli et al, 2024).

Acellular MPE fluid, regardless of the primary cancer type, has been demonstrated to induce epithelial to mesenchymal transition and increase cancer cell stemness within cancer cell lines that do not usually display these characteristics, such as the breast adenocarcinoma cell line MCF-7 and the LUSC cell line A549 (Asciak et al, 2021). This was demonstrated to occur via activation of the PI3K/AKT/mTOR pathway, which was activated by vascular endothelial growth factor (VEGF) secreted by cancer cells within the MPE (Yin et al, 2016).

Molecules such as VEGFa are up to 30-fold more abundant in MPE compared to normal pleural fluid. Furthermore, inhibition of vasoactive molecules (such as VEGF or TNF-α) has been shown to reduce vascular permeability and MPE fluid accumulation in vivo (Stathopoulos et al, 2008; Marazioti and Stathopoulos, 2014; Zebrowski et al, 1999). Several studies have reported that combination therapies, consisting of a chemotherapeutic agent (e.g., cisplatin, paclitaxel, gemcitabine) with anti-angiogenic agents

such as bevacizumab (a recombinant human monoclonal VEGF antibody) or endostar (a recombinant human endostatin, an anti-angiogenic protein) (Zhao et al, 2014; Dong et al, 2024), are significantly more effective in treating patients with MPE compared to chemotherapy alone. Despite the early recognition of these molecules as contributors to MPE formation (Xiang et al, 2022; Hooper et al, 2012; Kalomenidis et al, 2007; Psallidas et al, 2013), therapeutic targets blocking them are still not traditionally used in MPE patient populations outside of clinical trials.

## Outlooks for the treatment of MPE

Treatment options for patients diagnosed with MPE remain largely palliative and do not target their underlying pathophysiology. There remains no "one-size-fits-all" approach in treating these patients, with the most common treatments for MPE focused on interventions that prevent fluid accumulation and/or systemic anti-tumor therapy. Surgical intervention is not typically performed for MPE, as it represents stage IV cancer; however, in some cases, this approach may be employed to lyse pleural adhesions or locules, improve drainage, and/or promote pleurodesis to decrease fluid accumulation. However, this approach does little to address the residual malignant cell burden (especially in the case of MPE with low levels of solid tumor metastases), requiring combined drug treatment to improve patient outcomes (Disselhorst and Baas, 2020). Therefore, novel therapeutic options for MPE are urgently needed to improve upon the largely palliative treatment options currently available.

Recently, "thermal medicine" (heat) has been leveraged as a relatively simple way to modulate immune involvement, mimicking the body's natural immune response of fever production during illness and inflammation (Kozłowski et al, 2025; Wilander and Rathmell, 2025; Atencio et al, 2025; Evans et al, 2015). Fever-range hyperthermia (~41–45 °C) increases the efficacy of infused agents by transiently disrupting tight and adherens junctions, enhancing membrane and paracellular permeability, increasing vascular permeability, and lowering interstitial fluid pressure—together enabling deeper penetration into densely packed pleural tumor layers (van Stein et al, 2021; Dunne et al, 2020; Li et al, 2013). When using this in the context of MPE, both the lavage saline and intrapleural chemotherapeutic agents are routinely heated to ~43 °C (Hu et al, 2017; Sakaguchi et al, 2017), and this technique, termed hyperthermic intrathoracic chemotherapy (HITHOC) or intrapleural hyperthermic chemotherapy (IPHC), has demonstrated superior efficacy over systemic therapy alone and over normothermic intrapleural approaches (Järvinen et al, 2021; Zhou et al, 2017). Multicenter randomized and prospective studies showing higher objective response rates, longer progression-free survival, and improved overall survival (e.g., ORR 80.7% with HITHOC vs 31.0% with conventional intrapleural chemotherapy (Liu et al, 2023a) and VATS-assisted IPHC yielding shorter drainage duration, higher remission rates, and longer PFS than normothermic chemoperfusion (Cao et al, 2021). Preclinical and translational data further support enhanced cisplatin uptake, increased apoptosis, and reduced EGFR expression under intra-pleural hyperthermia, consistent with improved drug access (Shin et al, 2024). More recent reviews and clinical reports emphasize that combining hyperthermia with immune checkpoint inhibitors

(anti-PD-1–PD-1/PD-L1, anti-CTLA-4) enhances antitumor immunity and improves responses in refractory disease by boosting antigen presentation, T-cell infiltration, and immunogenic cell death (Abreu et al, 2025; Logghe et al, 2024; Yang et al, 2022). For MPE relevance, intrapleural perfusion hyperthermia has been shown to augment anti-PD-1 efficacy in lung adenocarcinoma with pleural involvement, underscoring the translational promise of hyperthermia-ICI combinations in pleural malignancy (Wang et al, 2024b; Tu et al, 2025). Taken together, these data support continued, standardized investigation of thermal medicine, optimizing temperature, duration, and combination strategies with immunotherapy and/or chemotherapy, as a means to improve local control and survival in MPE.

Additionally, the anti-cancer efficacy of immunomodulatory therapies has been well-established, including the administration of cGAMP, an activator of the STING pathway, particularly when delivered intrapleurally as liposomal cyclic-dinucleotide nanoparticles (LNP-CDN) (Liu et al, 2022). This pathway plays a protective role in cellular defense by activating innate and adaptive immunity in response to infection, tissue damage, or cell stress; it induces type I IFN and pro-inflammatory cytokines, enhances antigen presentation and DC maturation, and recruits CTLs, thereby converting "cold" to "hot" tumor microenvironments (Shen et al, 2025; Wu et al, 2025c; Yue et al, 2025; Pan et al, 2023). Intrapleural injection of lipid nanoparticles containing cGAMP (LNP-CDN designed to activate STING and type I-IFN signaling) has shown potent immunomodulatory effects and antitumor effects in experimental models of MPE by robustly activating STING in pleural macrophages and dendritic cells, reprogramming the MPE milieu from immune-suppressive to immune-active, inhibiting tumor growth in pleural and lung compartments, and prolonging survival; it also synergizes with anti-PD-L1 to reduce MPE volume, with congruent immune activation observed ex vivo in human MPE samples. Significant changes in traditionally pro-tumorigenic MPE resident immune cells were observed upon treatment, reduction of tumor-promoting myeloid populations, and mitigation of local immuno-suppression, culminating in anti-tumor activity. Nanoparticle-based intrapleural delivery also helps overcome pharmacokinetic barriers and enhance cellular uptake, supporting rational combination with immune checkpoint inhibitors for greater efficacy (Chen et al, 2025b; Guo et al, 2025; Xia et al, 2025).

## Conclusion

In this review article, we highlight key concepts that we believe are crucial to the continued understanding of MPE biology (Box 1). While significant advances in MPE biology have been made over the past 20 years, gaps in understanding remain regarding the factors that allow for the establishment and maintenance of metastatic pleural disease (Box 2). We are optimistic that a more thorough understanding of MPE pathophysiology will rapidly translate into better clinical therapeutic options and consequently result in improved outcomes among patients diagnosed with MPE.

### Pending issues in MPE research

Unanswered questions about MPE in lung cancer span basic biology and everyday clinical decisions. We still do not have a clear

> **Box 2   Current questions in MPE biology**
>
> 1. What factors predispose patients to MPE formation?
> 2. What causes more aggressive MPE with worsened prognosis, even when comparing the same primary cancer subtypes?
> 3. What drives metastasis to the pleural cavity over other sites (i.e., lymph nodes or liver)?
> 4. What role is played by neutrophils (and potentially NETs) within MPE formation?
> 5. Does the presence of MPE indicate more highly metastatic disease as compared to patients with dry pleural malignancy and/or cancers without effusions?
> 6. How can we develop more effective treatment protocols for patients with MPE using the therapeutic tools currently available to us? What therapeutic combinations can safely target multiple drivers of MPE?

map of how tumor, stromal, and immune cells work together to drive fluid buildup and local immune dysfunction, or which genomic and immune features of pleural metastases truly mark patients at the highest risk. Pleural fluid biomarkers and molecular assays such as ctDNA and miRNAs, alone or in multimarker panels, need prospective validation with practical cutoffs before they can reliably guide diagnosis or treatment. On the treatment side, trials focused on MPE itself are necessary to test the optimal combination of immunotherapy, anti-angiogenic drugs, and intrapleural approaches.

# Peer review information

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

## Acknowledgements

CVP was supported in part by the National Institutes of Health (NIH) R01CA215075, 1R01CA279532, 1R41CA246848, and 1R44-CA284932, a UCRF Innovator Award, Kickstarter Venture Services Commercialization awards, Lung Cancer Initiative of North Carolina Innovation and Alumni Awards, and a North Carolina Biotechnology Translation Research Grant (NCBC TRG). BGV and JA were supported in part by the UNC Lineberger Comprehensive Cancer Center University Cancer Research Fund. AAM was supported in part by the NIH NIGMS T32GM086330, NCI T32CA285257, Lung Cancer Initiative of North Carolina Research Fellowship, and the Robert A. Winn Excellence in Clinical Trials Career Development Award.

## Author contributions

**Allison T Woods**: Conceptualization; Writing—original draft; Writing—review and editing. **Abner A Murray**: Investigation; Writing—original draft; Writing—review and editing. **Benjamin G Vincent**: Conceptualization; Funding acquisition; Writing—original draft; Writing—review and editing. **Jason Akulian**: Conceptualization; Funding acquisition; Writing—original draft; Writing—review and editing. **Chad V Pecot**: Conceptualization; Funding acquisition; Writing—original draft; Project administration; Writing—review and editing.

## Disclosure and competing interests statement

CVP is a founder and shareholder in EnFuego Therapeutics. BGV is a co-founder and shareholder in Pathfinder Oncology and ShieldBreak Bio. AW, AAM, and JA declare no conflicts of interest.

