## [Peer Review File · EMBO Molecular Medicine]

Pathobiology and clinical significance of Malignant Pleural Effusions

Allison Woods, Abner Murray, Benjamin Vincent, Jason Akulian, and Chad Pecot

Corresponding authors: Chad Pecot (pecot@email.unc.edu) , Benjamin Vincent (benjamin_vincent@med.unc.edu), Jason Akulian (akulian@email.unc.edu)

Review Timeline:

Submission Date:	27th Oct 25
Editorial Decision:	29th Oct 25
Revision Received:	10th Nov 25
Accepted:	13th Nov 25

Editor: Zeljko Durdevic

Transaction Report:

Please note that the manuscript was previously reviewed at another journal. As EMBO Press has a transfer agreement with that journal, revision was invited based on the reports from that journal.

Reviewer Comments:

1. The title of this manuscript is "Cancer Biology of Malignant Pleural Effusions." The objective is to discuss and summarize tumor cells, their interactions with immune cells, and their roles in malignant pleural effusion (MPE) formation, while outlining clinical opportunities and promising research directions in MPE studies. The aim is to enhance understanding of MPE and improve prognosis for advanced cancer patients with MPE. However, the title does not sufficiently align with the manuscript's content.

We appreciate the reviewer's concern regarding alignment between the manuscript title and its content. We have carefully considered this feedback and, while maintaining the title "*Cancer Biology of Malignant Pleural Effusions*" to emphasize the central theme of tumor-driven biology in MPE pathogenesis, we have substantially broadened and updated the scope of the manuscript to fully align with the objectives outlined by the reviewer.

Specifically:

Integration of New Technologies: We now incorporate findings from single-cell RNA sequencing (scRNA-seq), spatial transcriptomics, and multiplex immunofluorescence studies that resolve TAM subclusters (e.g., IFN-TAMs vs. LA-TAMs), characterize T cell exhaustion programs, and stratify adaptive/innate immune landscapes in MPE. These additions provide state-of-the-art insights into the pleural tumor-immune microenvironment (TIME).

Expanded Tumor-Intrinsic Programs: We broadened the discussion of tumor cell-intrinsic drivers beyond canonical oncogenes (KRAS, EGFR, ALK, BRAF, MET, TP53, PI3K) to include metabolic and epigenetic reprogramming, tumor-derived cytokines and immunomodulators (IL-6, IL-8, TNF- α , TGF- β , osteopontin, CSF1), soluble checkpoints, integrins, and exosomal mediators (e.g., LRG1). We also added new subsections on discordant genomics and spatial heterogeneity in MPE, including differences between LUAD, squamous, and small cell variants, and tumor-driven spatial patterning of T cell proximity.

Deepened Immune Cell Coverage: We significantly expanded coverage of both innate and adaptive immune populations. Innate immunity now includes detailed subsections on TAMs (ontogeny, polarization, metabolic reprogramming, adrenergic signaling, IL-10→S100A9 axis), mast cells (CCL2/osteopontin-driven degranulation), neutrophils (CXCL1/2/8 recruitment, NET formation, prognostic role of NLR), dendritic cells, NK/ILCs, MDSC-like cells, eosinophils, basophils, and complement. Adaptive immunity now covers B cells, cytotoxic CD8⁺ T cells (exhaustion and impaired cytotoxicity), regulatory T cells (recruitment and suppression loops), and CD4⁺ helper subsets. We also discuss immune landscape stratification into "hot" and "cold" MPE clusters and their prognostic implications.

Non-Cellular and Acellular Drivers: We expanded this section to include vascular permeability (VEGF, TNF- α , MMPs), lymphatic obstruction, angiogenesis, extracellular mediators such as HMGB1 and surfactant proteins, and their cooperative role with tumor-immune crosstalk in sustaining MPE.

Diagnostic and Immunophenotyping Resource: A new subsection highlights the role of MPE as a liquid biopsy resource, enabling genomic profiling, immune landscape

stratification, and detection of therapeutically actionable discordance compared to matched tissue. This emphasizes both mechanistic insight and diagnostic potential.

Clinical Phenotypes – Dry vs. Wet Dissemination: We added discussion of “dry” versus “wet” pleural dissemination phenotypes, including their striking differences in survival outcomes, highlighting an underappreciated clinical distinction with implications for tumor–immune biology.

Improved Integration and Transitions: We revised section transitions to better link oncogenic drivers with their downstream effects on immune remodeling, directly addressing prior concerns about abrupt shifts in the narrative.

Updated References: We updated and expanded the reference base with extensive inclusion of 2019–2025 literature, ensuring comprehensive coverage of recent advances (e.g., PMID: 37604643, 39664577, 38951159, 38952674, 39721754). References were also reformatted to meet EMBO Press style.

Clinical and Translational Outlooks: We strengthened the translational aspects of the review by highlighting diagnostic opportunities in MPE sampling, discordant genomics in matched tumor vs. pleural fluid, and therapeutic strategies including STING agonists, CSF1R blockade, mast cell inhibition, β -blockers, checkpoint inhibitors, and intrapleural hyperthermic chemotherapy. The *Outlooks* section has been restructured to directly bridge mechanistic insights with therapeutic opportunities.

These revisions substantially broaden the scope of the manuscript beyond tumor-intrinsic mechanisms to encompass immune, acellular, and clinical dimensions of MPE biology. By integrating the latest single-cell and spatial immunogenomic studies (2023–2025), adding coverage of immune and non-cellular drivers, expanding clinical phenotypes, and strengthening translational perspectives, we ensure the manuscript now fully aligns with its title, *Cancer Biology of Malignant Pleural Effusions*.

We believe the updated structure and references now provide both a state-of-the-art reference and a forward-looking guide to advance research and improve prognosis for patients with MPE.

2. Line 61-62 : Ultimately, formation of effusions in malignant pleural disease is associated with worsened clinical outcomes. This statement lacks supporting references. Please provide data on the impact of MPE on survival outcomes in cancer patients.

We thank the reviewer for this insightful comment. In response, we have revised the text to include specific data on the impact of malignant pleural effusion (MPE) on clinical outcomes, as well as supporting references. The updated section (Lines 61–72) now includes median survival estimates for patients with MPE across multiple tumor types, highlights MPE as an independent adverse prognostic factor in NSCLC and cites key studies that quantify its association with reduced survival. These additions help contextualize the clinical significance of MPE and support the assertion that its presence is linked to worsened outcomes. Relevant references (Feller-Kopman & Light, 2018; Bielsa et al., 2008b; Morgensztern et al., 2012) have been added accordingly.

The Introduction should provide an overview of MPE; however, several important descriptions are missing.

We appreciate the reviewer's suggestion to provide a more comprehensive overview of malignant pleural effusion (MPE) in the Introduction. In response, we have significantly expanded this section to include:

- Epidemiologic data on MPE incidence across tumor types (e.g., NSCLC, breast, GI, hematologic malignancies).
- A more detailed description of clinical presentation, diagnostic modalities (cytology, imaging, biopsy), and associated survival outcomes.
- An overview of key pathogenic mechanisms including tumor-induced vascular hyperpermeability, impaired lymphatic drainage, and immune-tumor interactions.
- Clarification that although oncogenic mutations are associated with tumor biology, they are not currently understood to be the primary drivers of MPE formation.

These revisions provide a more complete and clinically grounded foundation for the review. We thank the reviewer for prompting this important improvement.

3. Line 63-65 : As such, defining the primary drivers of this disease complication and how these processes differ in MPEs compared to benign pleural effusion and/or primary tumors remains an ongoing area of investigation. This statement is redundant. The primary drivers are known (e.g., tumor biology for MPE vs. infection for benign effusions).

We appreciate the reviewer's observation. In response to this comment, we have removed the sentence, "As such, defining the primary drivers of this disease complication and how these processes differ in MPEs compared to benign pleural effusion and/or primary tumors remains an ongoing area of investigation." We agree that the primary etiologies of malignant and benign pleural effusions are well characterized, and the revised text instead focuses on current research efforts to delineate the specific molecular and cellular mechanisms—particularly those involving acellular components—that influence MPE biology and prognosis.

4. Line 67-70: Generally, MPE has been associated with mutations of common cancer-associated genes, modulation of the tumor immune microenvironment (TIME), increased vascular leakage and angiogenesis within the pleural space, and disturbances that reduce lymphatic pleural fluid drainage. MPE pathogenesis is primarily driven by tumor-induced vascular leakage, angiogenesis, and impaired lymphatic drainage - not necessarily by specific mutations. Current evidence shows limited knowledge about mutation and MPE development, with some mutations even exhibiting inverse associations.

We appreciate the reviewer's helpful clarification regarding the pathogenesis of MPE. In response, we have revised the text to more accurately reflect current understanding—emphasizing that tumor-induced vascular hyperpermeability, angiogenesis, and

impaired lymphatic drainage are the primary drivers of MPE formation. We also clarified that while oncogenic mutations may influence tumor biology, current evidence does not support them as central drivers of MPE pathogenesis and, in some cases, specific mutations may even show inverse associations with effusion development.

5. [H1] Cancer cell-intrinsic drivers of MPEs

As noted previously, this section should compare tumor cells from MPE(+) versus MPE(-) patients rather than describe common mutations. While mutational drivers are indeed complex and relevant, the authors should systematically review which specific gene mutations statistically associate with MPE formation (primary tumor vs. pleural mutations). Provide epidemiological evidence through comprehensive literature searching.

We thank the reviewer for this critical and constructive suggestion. In response, we have substantially revised the section to meet the request for a more rigorous and comparative analysis of MPE(+) versus MPE(-) tumors, rather than a general list of oncogenic mutations. The revised manuscript now provides:

1. **Epidemiologic comparisons of mutation prevalence:**

We incorporated recent genomic studies (e.g., Kang et al., Pineda et al., Ruan et al.) showing that EGFR mutations are enriched, while KRAS and STK11 mutations are depleted in MPE(+) NSCLC patients relative to MPE(-) controls. These population-level findings are contextualized against experimental model data to highlight concordance and divergence between human and preclinical systems.

2. **Mechanistic links to MPE biology:**

Rather than attributing MPE development directly to mutations, we now focus on how oncogenic mutations drive effusion via downstream effector pathways, including:

- KRAS → CCL2/CCR2, IL-1 β /NF- κ B, CD93/TEVs, eosinophilic CCL11 signaling
- EGFR → CXCR4/CXCL12 axis, IL-6/ANGPTL4 signaling, CD47 expression

These are supported by both mouse models and human MPE-derived tumor data.

3. **Transcriptional and spatial adaptations in MPE:**

We detail how pleural tumor cells diverge from their matched primary tumors, showing pleural-specific gene expression programs (e.g., IL-6, TGF- β , ANGPTL4, PD-L1), as demonstrated through paired scRNA-seq and spatial transcriptomics (Zhang et al., Mahmood et al., Temel & Derdiyok, Chang et al.).

4. **Molecular distinctions in ‘wet’ vs. ‘dry’ pleural disease:**

The revised text explicitly addresses the underexplored area of wet pleural dissemination (WPD) versus dry pleural dissemination (DPD), citing survival differences and emphasizing the need for further genomic characterization of these clinically distinct phenotypes (e.g., Kim et al., Shim et al.).

5. **Integration of real-world cfDNA evidence:**

We include recent data showing that MPE-derived cfDNA can reveal mutations not captured in tissue biopsies, with detection of EGFR, MET exon 14 skipping, and other drivers in real-world LUAD MPE cohorts. This reinforces the clinical and research utility of MPE sampling.

6. Framing of oncogenes within tumor–microenvironment interplay:

Rather than attributing MPE to mutations alone, we emphasize that MPE results from tumor-intrinsic transcriptional reprogramming in response to the pleural niche, supported by robust immune remodeling, metabolic adaptation, and matrix remodeling pathways.

In summary, the revised section directly addresses the reviewer’s concerns by replacing broad assertions with a systematic, mechanistic, and epidemiologically grounded analysis of MPE tumor biology. We believe this significantly improves the depth, accuracy, and translational relevance of the manuscript.

6. Line 87-90: The most common oncogenic drivers of LUAD have also been implicated in MPE development, including KRAS40,60, EGFR71–75, ALK71, BRAF76, MET77, TP5377, and PI3K54 among others. Despite recognition of these genes’ involvement in MPEs, whether these oncogenes are causally related to MPE remains unclear.

We appreciate the reviewer’s observation regarding the causal relevance of oncogenic drivers in MPE formation. In the revised manuscript, we have addressed this concern by clarifying the distinction between correlation and causality in the context of oncogenic mutations. Specifically:

- We now acknowledge that while mutations in genes such as KRAS and EGFR are frequently observed in LUAD and implicated in effusion biology, current evidence does not support a direct causal relationship between these mutations and MPE formation in human tumors.
- We reference clinical data showing that KRAS mutations are less frequent in MPE-positive NSCLCs, while EGFR mutations are more common—suggesting divergent associations that caution against oversimplified causal claims.
- The section has been reframed to emphasize that oncogenic drivers may contribute to MPE pathogenesis indirectly, through modulation of chemokine/cytokine signaling, immune remodeling, and metabolic adaptation, rather than acting as singular causal determinants.
- This is supported by updated references and the integration of transcriptomic, spatial, and single-cell profiling data that highlight pleural-specific adaptations distinct from the primary tumor genotype.

These clarifications are incorporated throughout the updated subsections under “Oncogenic Drivers and Chemokine Signaling,” particularly in the opening paragraph and the discussions of KRAS and EGFR. We thank the reviewer for prompting this important refinement in framing.

7. More concerns about these paragraphs. For example, KRAS. References (40 and 60) only demonstrate KRAS-driven MPE mechanisms in mice, lacking human clinical relevance. Notably, several human clinical studies report higher prevalence of KRAS

wild-type (vs. mutant) in MPE tumor cells (e.g., *Cell Oncol.* 35:189-196, OR 0.35 [0.14-0.86]). Authors should reconcile these contradictory findings.

We thank the reviewer for raising this important point. We have revised the KRAS subsection to directly address the lack of human clinical evidence for KRAS-driven MPE and to reconcile the apparent discrepancy between preclinical and clinical data. Specifically, we now emphasize that while mechanistic studies in mouse models demonstrate a KRAS–CCL2/CCR2–IL-1 β –NF κ B axis driving MPE formation, multiple human genomic datasets consistently report a lower prevalence of KRAS mutations in MPE-positive tumors compared with MPE-negative NSCLC (e.g., *Cell Oncol.* 35:189–196, OR 0.35 [0.14–0.86]). We attribute this divergence to differences between engineered mouse systems and the complex tumor–immune dynamics in human pleural disease. Accordingly, the revised text clarifies that current human clinical evidence does not support KRAS as a primary driver of MPE pathogenesis, and instead highlights other genetic alterations (e.g., EGFR enrichment, STK11 depletion) and pleural-specific transcriptional reprogramming as more consistent contributors. We also underscore the need for future multivariate and prospective studies to better define the role of KRAS in human MPE.

8. This section systematically reviews key driver mutations (KRAS and EGFR) and related downstream signaling alterations (e.g., CCL2-CCR2, NF- κ B, CXCL12-CXCR4) in MPE. Although other drivers such as MET, TP53, and PIK3CA, are less studied, they should still be briefly summarized.

We thank the reviewer for pointing out the need to briefly summarize additional oncogenic drivers beyond KRAS and EGFR. In the revised manuscript, we have expanded this section to include discussion of MET, TP53, PIK3CA, BRAF, and ALK, citing both preclinical and clinical studies. For example, we highlight preliminary evidence for MET overexpression, TP53 loss-of-function, BRAF V600E mutations, and PI3K/AKT/mTOR pathway activation as potential contributors to pleural invasion, immune modulation, or EMT. We also incorporated recent cfDNA sequencing data (Chang et al, 2024) demonstrating the detection of actionable mutations (EGFR, KRAS, MET, ALK, ERBB2) in MPE supernatants, supporting the clinical relevance of these alterations. Furthermore, we added discussion of histologic and co-mutation contexts (e.g., STK11/KEAP1 in squamous NSCLC, TP53/RB1 loss in SCLC), as well as pleural cytokine drivers such as TGF- β and osteopontin. Together, these additions ensure the review systematically covers both well-established and less-studied genetic drivers of MPE while highlighting key gaps for future research.

9. [H1] MPEs represent a highly complex TIME

As previously noted, this section suffers from significant omissions of recent advances. A comprehensive review of all immune cell populations in MPE is required, which the authors have failed to provide.

We appreciate the reviewer's feedback highlighting the need for a more comprehensive review of immune cell populations in MPE. In response, we have substantially revised and expanded the relevant sections to address these omissions.

Specifically, the subsection "*MPEs represent a highly complex TIME*" has been updated to better reflect recent advances in the field. Beyond the prior overview, the revised text now:

- Integrates insights from single-cell and transcriptomic profiling that detail immune heterogeneity in MPE.
- Expands discussion of immune landscape stratification (C1–C4 groups), including differential outcomes for adaptive+/innate+ versus adaptive-/innate- profiles.
- Links immune composition directly to survival outcomes, with explicit mention of poorer prognosis in C4 (adaptive-/innate+) groups and longer survival in C2 (adaptive+/innate-).
- Incorporates molecular signatures tied to prognosis, noting upregulation of PD-L1, B7-H3, VEGFA, and TREM2 in short-survival patients versus enrichment of pattern recognition receptor and type I interferon pathways in long-survival patients.
- Frames tumor-predominant vs. immune-predominant effusions within this stratification, providing a clearer link between cellular composition and clinical heterogeneity.

In addition, the *Innate Immune Cells in MPEs* section has been extensively developed. Whereas the earlier draft primarily focused on macrophages and briefly mentioned neutrophils and mast cells, the revised version now systematically reviews a much broader range of innate immune populations. This includes detailed coverage of tumor-associated macrophage subsets, mast cells, neutrophils and NETosis, dendritic cells, NK and innate lymphoid cells, MDSC-like populations, and contributions of less frequently discussed subsets such as eosinophils, basophils, and complement. Each subsection incorporates recent mechanistic and clinical findings, including studies linking immune subsets to pleural fluid dynamics, tumor-immune crosstalk, and patient survival outcomes.

10. Line 151-154: Key immunologic subsets thus far known to be integral for MPE formation include T cells, mast cells, tumor associated macrophages (TAMs), neutrophils, and myeloid derived suppressor cells (MDSCs) that infiltrate into the pleural space.^{81,82}

'Key immunologic subsets' remains undefined. While reference 81 is unavailable, reference 82 provides single-cell characterization of MPE immune landscapes. Notably, B cells-demonstrated to functionally contribute to MPE pathogenesis and correlate with MPE-mice survival outcomes (PMID: 29847991) - their exclusion requires explanation.

We thank the reviewer for this important observation. In the initial draft, the phrase "key immunologic subsets" was vague, relied on inconsistent references, and omitted B cells despite their demonstrated role in MPE pathogenesis and experimental models (e.g., PMID: 29847991). In the revised manuscript, we have addressed this concern by:

1. Removing the ambiguous line listing “key immunologic subsets,” which lacked sufficient clarity and specificity.
2. Correcting the references to cite recent studies that more accurately describe the immune composition of MPE, including single-cell characterizations (Huang et al, 2021b; Tumino et al, 2019; Wu et al, 2018).
3. Expanding our coverage of immune diversity in later sections (See comment 9). The updated text now emphasizes that both innate and adaptive cells—including B cells—participate in shaping the MPE tumor–immune microenvironment, with implications for prognosis and therapeutic responsiveness.

11. Line 163-194: This section discusses hot/cold tumors and immunotherapy, it lacks relevance to MPE. Critical questions remain unaddressed: (1) Do cold tumors preferentially develop MPE? (2) Not all immunotherapies show well efficacy against MPE, such as single PD-1/PD-L1 intrapleural injection. Because MPE is a cancer complication, any survival benefit likely reflects systemic tumor control rather than direct MPE modulation We recommend deletion, as this section lacks mechanistic connection to MPE pathogenesis.

We thank the reviewer for raising this important concern. We agree that our original draft overemphasized generic hot/cold tumor classifications and their implications for systemic immunotherapy, without adequately grounding the discussion in MPE-specific biology.

To address this:

1. Refocusing on MPE-specific findings: In the revised manuscript, we removed broad descriptions of hot/cold tumors and instead focused narrowly on the *MPE context*. The section now specifically reviews the immune profiling study by Wu et al. (2022), which stratified LUAD-MPEs into four immune landscape clusters (C1–C4). This framing highlights how adaptive and innate immune compositions correlate with survival in MPE patients, rather than implying that hot/cold tumors broadly dictate MPE biology.
2. Clarifying limitations: We explicitly state that there is no evidence that “cold” tumors preferentially develop MPE, nor that immune landscape alone determines effusion formation. Similarly, we note that the predictive value of hot/cold status for ICI responsiveness in MPE is unproven, and that survival benefits from ICIs are most likely due to systemic tumor control rather than direct modulation of the effusion.
3. Establishing mechanistic linkage: The section now serves as a transitional framework to set up the subsequent, detailed discussion of individual immune cell subsets in MPE (TAMs, mast cells, neutrophils, B cells, T cells, etc.). This grounds the prognostic observations in cellular mechanisms, addressing the reviewer’s concern that the original text lacked a mechanistic connection to MPE pathogenesis.

We believe these revisions successfully eliminate irrelevant discussion, place the focus on effusion-specific data, and clarify both the value and limitations of immune landscape profiling in MPE.

Line 163-165: Over the past 10 years, stratification systems have been identified to broadly categorize cancers into either “hot” or “cold” status based on their immune landscape and predicted response to immune checkpoint inhibitors (ICI). Please provide references.

We thank the reviewer for this comment. The sentence referencing generic hot/cold tumor stratification and its application to immunotherapy has been removed in the revised draft.

12. [H2] Non-Cellular drivers of MPE

It's known that many cytokines and chemokines play vital roles in MPE pathogenesis. Because these soluble proteins are secreted by tumor cells or other cells, this part should be deleted and properly integrated into the tumor cell or TIM section. As a standalone section, a systematic description is required rather than mention of several select proteins.

We thank the reviewer for pointing out the potential overlap between the “Non-Cellular Drivers of MPE” section and the tumor/TIME cytokine subsections. We have revised this portion of the manuscript to clearly delineate its scope:

- The “**Tumor-Derived Cytokines and Immune-Modulatory Factors**” section provides a systematic and detailed review of cytokine and chemokine biology in MPE, including IL-6, IL-8, TGF- β , TNF- α , CSF1, and others.
- By contrast, the “**Non-Cellular Drivers**” section is now explicitly framed around the *structural determinants of fluid homeostasis*, namely lymphatic obstruction and vascular hyperpermeability. Mentions of cytokines (e.g., VEGF, TNF- α) are retained only in brief to illustrate their direct role in altering vascular integrity and driving effusion accumulation, not to duplicate their tumor-intrinsic or immunomodulatory functions already covered in detail above.

13. [H2] Adaptive Immune Cells in MPEs

This section is overly simplistic and omits critical information. I recommend expanding the discussion of CD4+ subpopulations and their functional relationships, with reference to established literature (e.g. Helper T cells in malignant pleural effusion, *Cancer Letters*, 2021).

We thank the reviewer for this insightful comment. The “Adaptive Immune Cells in MPEs” section has been substantially expanded to address these concerns and now provides a comprehensive, mechanistic discussion of CD4+ T-cell subpopulations and their functional interplay. The revised text integrates new literature (including *Cancer Letters*, 2021) and recent studies (2020–2025) describing Th1, Th17, Th22, Treg, and tissue-resident memory (T_{rm}) subsets, their cytokine networks (IL-6, IL-23, IL-26, IL-10), and transcriptional or metabolic reprogramming within the pleural environment. We

added coverage of the IL-26–Th22 axis, IL-10/miRNA-mediated Th1 suppression, and microRNA-driven differentiation control. Single-cell and post-immunotherapy datasets were incorporated to illustrate CD4⁺ plasticity and exhaustion. Collectively, these revisions enrich the depth of adaptive immunity analysis, clarify subset relationships, and align the section with current immunologic understanding of MPE biology.

14. [H3] B cells in malignant pleural effusions

B cells have demonstrated prognostic significance in MPE mouse models, yet their functional role remains underexplored in this review. Notably, IL-10⁺ B cell subsets - known to modulate the MPE immune microenvironment - are conspicuously absent from the discussion. A comprehensive update on recent B-cell research in MPE pathogenesis is needed.

We agree and have comprehensively expanded the B-cell section. The revision now (i) explicitly covers IL-10⁺ regulatory B cells (Bregs; CD24^{hi}CD27⁺), including their enrichment in pleural fluid vs. blood, expression of BAFF-R, Galectin-1, CD39, and IL-10–mediated suppression of Th1 responses/Treg promotion; (ii) integrates single-cell maps that resolve multiple pleural B-cell and plasma-cell states and their hypoxia/glycolysis and interferon-responsive programs (↑STAT1/IRF9), as well as a naïve-skewed, plasma-depleted pleural composition; (iii) situates pleural B cells within the emerging pan-cancer B-cell taxonomy (e.g., TAABs vs. regulatory/naïve states) and notes the largely non-clonal BCR architecture in MPE; (iv) details mechanistic crosstalk—BAFF/TNFSF13B survival signals from mesothelial/myeloid cells, preferential Breg engagement of CD4⁺ T cells, and Th17/IL-21 reinforcement of B-cell activation/class switching; and (v) strengthens prognostic and preclinical links, including murine evidence of B-cell–driven effusion formation and human cohorts where higher B-cell infiltration/B-cell:leukocyte ratios (with low neutrophils) associate with improved survival (while acknowledging subset-specific effects).

15. Although the authors discuss innate immune cells, including tumor-associated macrophages, neutrophils and mast cells., and adaptive immune cells, including CD4⁺ T cells, Tregs, Cytotoxic T cells, and B cells, and majority of studies are cited, more recent advances on immune cells in MPE are missing, failing to reflect current research progress. The authors should update the knowledge and extensively review the literatures, and give a more comprehensive summarization. (e.g. Macrophages, PMID: 37604643; PMID: 39664577; Eosinophils, PMID: 38951159; CD8⁺ T cells, PMID: 38952674; Tregs, PMID: 39721754)

We thank the reviewer for this important suggestion. In the revised manuscript, we have substantially updated the sections on both innate and adaptive immune cells in MPE to incorporate the most recent advances, including the specific studies referenced (PMIDs: 37604643, 39664577, 38951159, 38952674, 39721754). These updates ensure inclusion of the newest insights into macrophage heterogeneity, eosinophil contributions, CD8⁺ T cell dysfunction, and regulatory T cell–mediated immunosuppression in MPE.

We also conducted a systematic update of the literature. The revised immune cell sections now cite **178 studies in total**, spanning 2004–2025, with a strong emphasis on the most recent work:

- 2019 → 14 references
- 2020 → 13 references
- 2021 → 19 references
- 2022 → 20 references
- 2023 → 24 references
- 2024 → 19 references
- 2025 → 14 references

This ensures that the review reflects not only the historical foundation of the field but also the rapid progress made over the last 3–4 years. We believe the revisions now provide the comprehensive and up-to-date overview requested by the reviewer.

16. It seems that there are insufficient chapter transitions; for example, the transition from “Intrinsic Drivers of Cancer Cells” to “Immune Microenvironment” appears abrupt. There is a lack of clear elucidation regarding how oncogenes (e.g., *KRAS*, *EGFR*) directly or indirectly regulate the functions of immune cells.

We thank the reviewer for highlighting the need for clearer transitions between the “Cancer cell-intrinsic drivers” and “Immune microenvironment” sections, and for emphasizing the importance of explicitly linking oncogenic drivers with immune modulation in MPE. We have revised the section to strengthen these transitions and highlight mechanistic connections.

Specifically:

- At the close of the oncogenic drivers subsection, we now include bridging text that emphasizes how oncogenes such as *KRAS* and *EGFR* not only drive tumor cell-intrinsic programs but also profoundly re-shape the pleural immune landscape. For example, *KRAS*-mutant tumors drive *CCL2* secretion and recruitment of *CCR2*⁺ myeloid cells, and *EGFR*-mutant tumors remodel the TIME via *CXCL12*–*CXCR4* signaling to promote neutrophil enrichment. These mechanisms illustrate the direct interface between oncogenic signaling and immune cell function.
- We clarified that these pathways provide the mechanistic basis for immune composition differences in MPEs (e.g., neutrophil-predominant vs lymphocyte-predominant effusions), thereby setting up the transition into the “MPEs represent a highly complex TIME” section.
- The revised transition text explicitly states that tumor-intrinsic signaling programs establish the cytokine, chemokine, and metabolic conditions that shape the immune microenvironment, thereby linking the two major sections more clearly.

Together, these changes ensure that the narrative flows logically from tumor cell-intrinsic drivers to the immunologic composition of the MPE, while also highlighting mechanistic connections between oncogenes and immune subsets.

17. More and more new technologies have been applied to the study of the mechanism of MPE, such as single cell RNA sequencing. These research results should also be included in this review and directly described and discussed.

We appreciate the reviewer's suggestion to incorporate discussion of recent technological advances in MPE research, particularly single-cell RNA sequencing (scRNA-seq). In the revised manuscript, we have directly integrated scRNA-seq and related high-resolution profiling studies throughout both the cancer cell-intrinsic and immune cell sections. For example, we now highlight that scRNA-seq stratified LUAD-MPEs into four immune landscape clusters (C1–C4), showing that “hot” adaptive clusters correlated with improved survival, whereas innate-dominant “cold” clusters correlated with poor prognosis.

We further detail how scRNA-seq and spatial analyses have defined TAM ontogeny and end-states (IFN-TAMs vs. lipid-associated LA-TAMs), revealed their metabolic reprogramming (cholesterol, iron-handling, complement), and identified ligand–receptor networks (e.g., VEGFA/MIF→SPP1⁺/CD74⁺ TAMs) that drive vascular permeability and immunosuppression. Similarly, we discuss how high-dimensional profiling has illuminated mast-cell recruitment (CCL2, OPN), neutrophil chemoattraction (CXCL1/2/8), and emerging therapeutic targets such as β -adrenergic metabolic programming in macrophages and intrapleural STING agonism.

Together, these additions ensure that the review now explicitly describes how single-cell RNA sequencing, secretomics, and spatial multi-omics have transformed our understanding of both adaptive and innate immune compartments in MPE. These technologies are emphasized as critical in revealing the heterogeneity, functional states, and therapeutic vulnerabilities of the pleural TIME.

18. The figure legends to figures 2 and 3 are missing, and the related information described in the last version is insufficient; the figure legends should describe more detailed information to illustrate the figure contents.

We thank the reviewer for this valuable suggestion. In the revised manuscript, the legend to Figure 2 has been substantially expanded to provide a more detailed and mechanistically oriented description of the figure contents. The new legend now outlines the contributions of each major cellular and non-cellular driver of malignant pleural effusion (MPE) pathogenesis, explicitly linking them to their depicted roles in the schematic.

We appreciate the reviewer's suggestion and have substantially expanded the legend for Figure 3 to provide a more detailed description of the cellular and molecular interactions depicted. The revised caption now clearly outlines the mechanisms by which tumor cells polarize immune subsets toward pro-tumorigenic functions within the pleural space. Specifically, the legend describes tumor-derived CCL2–CCR2 signaling that recruits macrophages and mast cells, HMGB1-mediated inflammasome activation driving IL-1 β release from TAMs, osteopontin-dependent mast cell degranulation

releasing tryptase and IL-1 β , neutrophil recruitment and NET formation, and T cell exhaustion with enrichment of immunosuppressive Tregs. These details align directly with the figure contents and integrate mechanisms described in the tumor-intrinsic and immune TIME sections of the manuscript.

We believe this revision addresses the reviewer's concern by ensuring that the legend is not only descriptive but also mechanistically informative, thereby helping readers fully interpret the visual summary.

19. The references cited in this review and their placement in the article are confusing; please pay more attention to the order of the cited literature.

We thank the reviewer for pointing out the need for improved clarity and consistency in the use of references. In the revised manuscript, we have thoroughly reviewed and reorganized all citations to ensure they follow the correct chronological and topical order, and we have reformatted them to comply with EMBO Press style guidelines. References have been cross-checked against the text to ensure accurate placement within each section, with particular attention to grouping foundational studies first, followed by more recent advances. This restructuring eliminates earlier inconsistencies and should greatly improve readability and transparency for the reader.

29th Oct 2025

Dear Dr. Pecot,

Thank you for the submission of your revised manuscript to EMBO Molecular Medicine. I am pleased to inform you that we will be able to accept your manuscript pending the following final amendments:

- 1) Considering the scope of our journal please consider changing the title of the manuscript to "Pathobiology and clinical significance of malignant pleural effusions".
- 2) Please submit a clean version of the manuscript without track changes. The designations [H1], [H2] and [H3] are not clear, particularly in cases where there are both [H2] and [H3] subheadings of a section. Please clarify and only leave the correct versions in the revised manuscripts.
- 3) Figures:
 - Please remove all figures from the manuscript and leave only their legends (without boxes) at the end of the file.
 - We have sent the figures for the final adjustments to our graphic designer, who will contact you for your approval.
- 4) Tables: Please move Table 1 to the main manuscript file and place it after figure legends. Incorporate table references into the main "References". In the table a reference should also be cited by author and year of publication. Please correct.
- 5) Add up to 5 keywords.
- 6) Remove "Abbreviations/acronyms".
- 7) Rename "Competing interests statement" to "Disclosure and competing interests statement". We updated our journal's competing interests policy in January 2022 and request authors to consider both actual and perceived competing interests. Please review the policy <https://www.embopress.org/competing-interests> and update your competing interests if necessary.
- 8) Move "Acknowledgments" and "Disclosure and competing interests statement" before "References".
- 9) Author contributions: Please remove it from the manuscript and specify author contributions in our submission system. CRediT has replaced the traditional author contributions section because it offers a systematic machine-readable author contributions format that allows for more effective research assessment. You are encouraged to use the free text boxes beneath each contributing author's name to add specific details on the author's contribution. More information is available in our guide to authors:
<https://www.embopress.org/page/journal/17574684/authorguide#authorshipguidelines>
- 10) Pending issues: At the end of each article is a box highlighting issues that still need further studies and where research efforts should converge. Could you identify some pending issues?
- 11) As part of the EMBO Publications transparent editorial process EMBO Molecular Medicine will publish online a Review Process File (RPF) to accompany accepted manuscripts. This file will be published in conjunction with your paper and will include your point-by-point response and all pertinent correspondence relating to the manuscript. Let us know whether you agree with the publication of the RPF.
- 12) Please provide a point-by-point response letter to my comments (as Word file).

I look forward to receiving the revised version of your manuscript.

Yours sincerely,

Zeljko Durdevic

Zeljko Durdevic
Senior Editor
EMBO Molecular Medicine

*** IMPORTANT INFORMATION ***

- 1) a .doc formatted version of the manuscript text (including Figure legends and tables)
- 2) Separate figure files
- 3) a letter INCLUDING the reviewer's reports and your detailed responses to their comments.

Also, and to save some time should your paper be accepted, please read below for additional information regarding some features of our research articles:

1) Glossary: EMBO Molecular Medicine articles will be accompanied by a glossary explaining some of the terms used for laymen. I identified the following:

_____, _____, _____

Could you please help us in identifying terms that may need an "explanation" other terms that we can add to the glossary.

2) Pending issues: At the end of each article we will have a box highlighting issues that still need further studies and where research efforts should converge (we call this the Pending issues box). From my reading I would say:

but I can see there may be many more. Could you work on this as well?

3) Disclosure and competing interest statement: Please include a statement declaring any competing commercial interests in relation to your submitted work.

4) Please note that we now mandate that all corresponding authors list an ORCID digital identifier. This takes <90 seconds to complete. We encourage all authors to supply an ORCID identifier, which will be linked to their name for unambiguous name identification.

Currently, our records indicate that the ORCID for your account is 0000-0002-5250-2999.

Link Not Available

-

Thank you,

Zeljko Durdevic

The authors addressed the remaining editorial issues.

13th Nov 2025

Dear Dr. Pecot,

We are pleased to inform you that your manuscript is accepted for publication and is now being sent to our publisher to be included in the next available issue of EMBO Molecular Medicine pending the final figure adjustments by our graphic designer.

Your manuscript will be processed for publication by EMBO Press. It will be copy edited and you will receive page proofs prior to publication. You will soon be contacted by Springer Nature to sign your publishing license. When you login to the customer service website, please use the following token to waive the article publication charges. Should you experience any difficulty, please email publishing@embo.org.

Waiver token: [removed]

Zeljko Durdevic
Senior Editor
EMBO Molecular Medicine